

# Aerosol influences on low-level clouds in the West African monsoon

Jonathan W. Taylor[1], Sophie L. Haslett[1,*], Keith Bower[1], Michael Flynn[1], Ian Crawford[1], James Dorsey[1,2], Tom Choularton[1], Paul J. Connolly[1], Valerian Hahn[3], Christiane Voigt[3,4], Daniel Sauer[3], Régis Dupuy[5], Joel Brito[5,**], Alfons Schwarzenboeck[5], Thierry Bourriane[6], Cyrielle Denjean[6], Phil Rosenberg[7], Cyrille Flamant[8], James D. Lee[9,10], Adam R. Vaughan[9], Peter G. Hill[11], Barbara Brooks[12], Valéry Catoire[13], Peter Knippertz[14], and Hugh Coe[1]

[1]Centre for Atmospheric Science, School of Earth and Environmental Sciences, University of Manchester, Manchester, UK
[2]National Centre for Atmospheric Science, University of Manchester, Manchester, UK
[3]Institut für Physik der Atmosphäre, Deutsches Zentrum für Luft- und Raumfahrt (DLR), Oberpfaffenhofen, Germany
[4]Institut für Physik der Atmosphäre, Johannes Gutenberg-Universität Mainz, Germany
[5]Laboratoire de Météorologie Physique, Université Clermont Auvergne, Clermont-Ferrand, France
[6]CNRM, Université de Toulouse, Météo-France, CNRS, Toulouse, France
[7]University of Leeds, Leeds, UK
[8]LATMOS/IPSL, Sorbonne Université, UVSQ, CNRS, Paris, France
[9]Wolfson Atmospheric Chemistry Laboratories, University of York, York, UK
[10]National Centre for Atmospheric Science, University of York, York, UK
[11]Department of Meteorology, University of Reading, Reading, UK
[12]National Centre for Atmospheric Science, Leeds, UK
[13]Laboratoire de Physique et Chimie de l'Environnement et de l'Espace (LPC2E), Université Orléans-CNRS-CNES, Orléans Cedex 2, France
[14]Institute of Meteorology and Climate Research, Karlsruhe Institute of Technology, Karlsruhe, Germany
*Now at: Department of Environmental Science and Analytical Chemistry, Stockholm University, Stockhkolm 114 18, Sweden
**Now at: IMT Lille Douai, Univ. Lille, SAGE, 59000 Lille France

**Correspondence:** Jonathan Taylor (jonathan.taylor@manchester.ac.uk)

**Abstract.** Low-level clouds (LLC) cover a wide area of southern West Africa (SWA) during the summer monsoon months, and have an important cooling effect on the regional climate. Previous studies of these clouds have focused on modelling and remote sensing via satellite. We present the first comprehensive set



of regional, in situ measurements of cloud microphysics, taken during June – July 2016, as part of the DACCIWA (Dynamics-Aerosol-Chemistry-Clouds Interactions in West Africa) campaign, assessing spatial and temporal variation in the properties of these clouds.

LLC developed overnight and mean cloud cover peaked a few hundred kilometres inland around 10:00 local solar time (LST), before clouds began to dissipate and convection intensified in the afternoon. Additional sea breeze clouds developed near the coast in the late morning, reaching a maximum extent around 12:00 LST. Regional variation in LLC cover was largely determined by the modulation of the cool maritime inflow by the local orography, with peaks on the upwind side of hills and minima on the leeward sides. In the broad-scale cloud field, no lasting impacts related to anthropogenic aerosol were observed downwind of major population centres.

The boundary layer cloud drop number concentration (CDNC) was locally variable inland, ranging from 200 to 840 $\text{cm}^{-3}$ (10th and 90th percentiles at standard temperature and pressure), but showed no systematic regional variations. Enhancements were seen in pollution plumes from the coastal cities, but were not statistically significant across the region. The majority of accumulation mode aerosols, and therefore cloud condensation nuclei, were from ubiquitous biomass burning smoke transported from the southern hemisphere. Consequently, all clouds measured (inland and offshore) had significantly higher CDNC and lower effective radius than clouds over the remote south Atlantic from literature.

A parcel model sensitivity analysis showed that doubling or halving local emissions only changed the calculated CDNC by 13 – 22%, as the high background meant local emissions were a small fraction of total aerosol. As the population of SWA grows, local emissions are expected to rise. Biomass burning smoke transported from the southern hemisphere is likely to dampen any effect of these increased local emissions on cloud-aerosol interactions. An integrative analysis between local pollution and Central African biomass burning emissions must be considered when predicting anthropogenic impacts on the regional cloud field during the West African monsoon.

# 1 Introduction

During the summer monsoon in June – September, large areas of southern West Africa (SWA) are covered by low-level clouds (LLC), which form overnight and thicken in the morning, before breaking up in the early



afternoon (van der Linden et al., 2015). By presenting a high albedo surface close to the ground, these clouds generate a strong surface cooling. Many climate models struggle to accurately represent LLC (Hannak et al., 2017), and measurements of LLC (and their radiative interactions with higher-level cloud layers) are a key uncertainty in the quantification of the overall cloud radiative effect in SWA (Hill et al., 2018). Most of

the population centres in SWA are located near the coast, and plumes of local anthropogenic pollution are transported inland (Deroubaix et al., 2018), potentially being entrained into cloud base. Over the coming decades, the population of SWA is expected to undergo large increases (United Nations, 2017), leading to corresponding increases in emissions of anthropogenic pollution (Liousse et al., 2014). Such increases may affect dynamics and cloud microphysics in the region, and it is therefore of interest to determine any impact

on the regional climate such as changes in cloud cover and precipitation (Knippertz et al., 2015b).

The DACCIWA (Dynamics-Aerosol-Chemistry-Clouds Interactions in West Africa) project (Knippertz et al., 2015a) was developed to provide a comprehensive overview of cloud-aerosol-precipitation interactions in the region. The program studied different scales, from local emission measurements near source, to regional sampling using aircraft, remote sensing and model analyses. This study focuses mostly on in

situ cloud measurements made during the DACCIWA aircraft campaign (Flamant et al., 2018b), which took place between 29 June and 16 July 2016. Three research aircraft, each equipped with a suite of atmospheric measurement probes, were based out of Lomé, Togo, and conducted 50 research sorties flying over Togo, Benin, Ghana and Côte d'Ivoire, of which 33 included in-situ sampling of cloud properties.

The flying campaign took place in typical, post-onset West African monsoon conditions (Knippertz et al.,

2017). In the daytime continental boundary layer, the southwesterly monsoon flow dominates the wind field, particularly in the lower kilometre of the atmosphere. Above around 1.5 – 2 km, the wind direction shifts to easterly, and some easterly waves and vortices passed through the region during the study period (Knippertz et al., 2017). Near the coast, winds are slowed by boundary layer turbulence, generating sea breeze clouds along a convergence front that moves up to a few tens of kilometres inland during the day (Adler et al., 2017;

Deetz et al., 2018b; Flamant et al., 2018a). As the turbulence subsides in the evening, the monsoon flow strengthens to bring this front inland as the nocturnal low-level jet (NLLJ), a strong flow of cool and humid maritime air, often penetrating over 180 km inland (e.g. Kalthoff et al., 2018). We follow the convention of Adler et al. (2018) in referring to the various flows bringing marine air inland as the Gulf of Guinea maritime inflow, or simply maritime inflow for short.





The NLLJ appears to be a key factor for initiating the formation of nocturnal LLCs, and previous studies have suggested several factors may be at play. The jet is strongest a few hundred metres above ground level (Kalthoff et al., 2018), and Schrage and Fink (2012) suggested shear-driven turbulence below this maximum draws moisture upwards from near the surface. Schuster et al. (2013) concluded that cloud formation was the

result of a subtle balance of several factors including shear-driven turbulence, orographic lift, and cooling by advection of cool air in the maritime inflow. Adler et al. (2018) recently quantified the boundary layer heat budget from 11 case studies using observations from a ground station near Savé during the DACCIWA campaign. They calculated that 50% of the cooling prior to LLC formation was due to advection of cold maritime air, with roughly 20% each from radiative and sensible heat flux divergence. A similar analysis by

Babić et al. (2018) on a single case study found advection of cool air was responsible for 72% of boundary layer cooling. The importance of the maritime inflow in cooling a particular location means that orography plays a significant role in determining geographical variation in the cloud field. Higher LLC coverage is found on the leeward side of slopes, where the maritime flow can reach more easily, and where this flow is also forced to rise orographically (van der Linden et al., 2015).

Deetz et al. (2018b) modelled the effects of changing aerosol emissions on the cloud fields in SWA. They suggested that the reduction of land-sea temperature gradient from increasingly hazy conditions led to a weakening of the monsoon flow and nocturnal low-level jet, and a delay in stratocumulus to cumulus transition and cloud breakup. Haslett et al. (submitted, 2019) recently described the aerosol properties measured during the DACCIWA aircraft campaign, and showed that a large background of transported biomass burning

pollution from the southern hemisphere was ubiquitous in the West African boundary layer. Although city emissions resulted in aerosol load increase directly downwind of urban conglomerates (Brito et al., 2018), the injection of aerosols prone to activate as cloud nuclei ($> 0.1$ μm) are thought to have limited impact on an already elevated background. This transported biomass burning background was reproducible in a modelling study by Menut et al. (2018). Painemal et al. (2014) showed that cloud-aerosol interactions with biomass

burning smoke were the main factor governing cloud microphysical properties in clouds north of 5°S over the South Atlantic, but so far no study has measured the microphysical properties of clouds over SWA to quantify the relative influences of transported pollution and local anthropogenic aerosol emissions.

The objectives of this paper are to:



1. Provide a statistical overview of in situ cloud properties measured during the DACCIWA aircraft campaign.

2. Compare the broad-scale cloud field during the DACCIWA aircraft campaign to previous overviews of the region, to assess the representativeness of the study region and period.

3. Assess the impacts of local urban emissions and transported background pollution on cloud properties in the region.

The conclusions we draw may be used in process and regional modelling studies, to improve assessments of the impacts of increasing urban emissions on regional cloud, and consequently precipitation and climate, under different pollution scenarios.

## 2   The DACCIWA aircraft project

The flying campaign took place from 29 June – 16 July 2016, and was a collaboration between science teams flying on three aircraft: the British Twin Otter operated by British Antarctic Survey, the French ATR42 (hereafter referred to as the ATR) operated by SAFIRE (Service des Avions Francais Instrumentés pour la Recherche en Environnement), and the German Falcon aircraft operated by DLR (Deutsches Zentrum für Luft- und Raumfahrt). The project has been described in detail (Flamant et al., 2018b); here we provide a brief description of the flights and relevant instrumentation. Research sorties were conducted in daylight hours from Lomé airport (6.17°N, 1.25°E), though the ATR refuelled twice in Abidjan (5.26°N, 3.93°W). Figure 1 shows flight tracks from all three aircraft below 1 km. The local time in Togo, Ghana and Côte D'Ivoire is UTC, though Benin uses UTC+1.

Many sorties were flown along a SW-NE axis from Lomé to Savé (as shown in Fig. 1), close to the average direction of the low-level monsoon flow. These flights included profiles up and down through cloud, as well as straight and level runs above, in, and below cloud to provide statistical mapping of clouds, aerosol and radiation. Other flight objectives included targeting measurements of emissions from different sources (such as cities, oil rigs, and ships), and mapping clouds and pollution further west over Ghana and Côte d'Ivoire. The large majority of measurements were taken over the African continent. Flights over the sea were carried out on several days during the campaign, and these are sufficient to gauge the variability of




aerosol offshore. Cloud penetrations on these offshore flights were relatively few, and it is difficult to gauge how statistically robust the offshore cloud measurements were. In this analysis we use cloud data from all flights of the campaign, and mostly only consider data from the lowest 1 km to investigate the effects of aerosol on boundary layer clouds over the African continent.

## 2.1 In situ instrumentation

Each aircraft was equipped with a suite of instrumentation to measure basic meteorological variables such as temperature, humidity, pressure, and winds. Further details are provided by Flamant et al. (2018b). Cloud drops $3 - 50$ μm in diameter were measured using a cloud droplet probe (CDP) on the Twin Otter, CDP and/or fast CDP on the ATR, and a cloud-aerosol spectrometer (CAS) on the Falcon (Baumgardner et al., 2001; Voigt et al., 2017). Both CDPs had modified pinholes to reduce coincidence (Lance, 2012), however the CAS did not, and the CAS CDNC measurements were corrected for coincidence at high concentrations as described by Kleine et al. (2018). Each instrument had its sample area measured using a droplet gun prior to the campaign, and sizing was calibrated in the field using glass beads of known size and refractive index. To ensure comparability between CDNC measurements on the different platforms, we performed a statistical analysis of all CDNC measurements made between 0 and 100 km inland over Togo/Benin at altitudes below 1 km. CDNC measured on the different platforms showed excellent agreement, with the medians and quartiles all agreeing within 5%.

We also consider cloud drop effective radius, defined as

$$R_{\text{Eff}} = \frac{\int_0^\infty R^3 \, n(R) \, dR}{\int_0^\infty R^2 n(R) dR},$$

where $n(R)$ is the number concentration of drop with radius $R$.

For $R_{\text{Eff}}$, average vertical profiles between 0 and 100 km inland showed the cloud probes agreed within ~1 μm, which is within the uncertainties of the instruments.

Data were considered in-cloud when the measured liquid water content (LWC) was greater than $0.1 \, \text{g m}^{-3}$. This relatively high LWC threshold minimises the effects of diffuse cloud edges on our measurements, and removes swollen aerosol layers (see Deetz et al., 2018a; Haslett et al., 2018). Measurements of CDNC are reported in number per cubic centimetre ($\text{cm}^{-3}$), corrected to standard temperature and pressure (STP, 273.15 K and 1013.25 hPa).



Aerosol composition was measured by compact time-of-flight aerosol mass spectrometers (AMS, Drewnick et al., 2005). Two AMS instruments were mounted on the ATR and Twin Otter, which were each calibrated using nebulised ammonium nitrate and ammonium sulfate to determine absolute and relative ionization efficiency. We also consider aerosol size distributions, which were measured using GRIMM optical particle counters (model 1.109 on the ATR and 1.129 on the Twin Otter and Falcon), and a scanning mobility particle sizer (SMPS) on board the ATR. Further details on the AMS and SMPS are provided by Brito et al. (2018) and Haslett et al. (submitted, 2019). From the GRIMM measurements we consider only the total concentration of particles larger than 250 nm ($N_{250}$), and from the SMPS we use aerosol size distributions in the range 20 – 500 nm. The strength of the GRIMM dataset is that all three aircraft had instruments running on every flight, making it a useful dataset for measuring changes in accumulation mode aerosol concentrations. The SMPS data coverage is more limited, and is restricted to straight and level runs, but provides a more detailed measurement of the aerosol size distribution. Aerosol data were screened for cloud using a threshold for LWC of 0.01 $\mathrm{g\,m^{-3}}$ and CDNC of 10 $\mathrm{cm^{-3}}$.

Carbon monoxide (CO) concentrations were measured using infrared absorption spectrometry, by an Aerolaser AL5002 on the Twin Otter, SPectrometre InfraRouge In situ Toute altitude (SPIRIT, Catoire et al., 2017) on the Falcon, and a Picarro Analyzer G2401-m on the ATR. We also consider vertical velocity, which was measured using wind probes on the ATR and Falcon.

### 2.1.1 Cloud satellite measurements

Using satellite measurements allows us to view the cloud field over a large region. We used the optimal cloud analysis (OCA, Watts et al., 2011) product taken from the Meteosat Spinning Enhanced Visible and InfraRed Imager (SEVIRI) spectrometer. This product provides cloud top pressure (CTP) and cloud optical thickness (COT) for the top two cloud layers (looking from above), for scans every 15 minutes. We use this product to derive the LLC fraction (for pressures above 680 hPa), as described in appendix A. Comparison with the LLC fraction using ceilometers, based at the DACCIWA supersites near Savé and Kumasi (Kalthoff et al., 2018), showed that this satellite-derived product agreed with ground measurements within ~10% cloud fraction, capturing both the absolute values and also the diurnal cycle, particularly during daylight hours. Further details are provided in appendix A.



### 2.1.2 Parcel modelling

Parcel model simulations of aerosol activation were carried out using the Aerosol-Cloud and Precipitation Interactions Model (ACPIM) (Connolly et al., 2009). Multiple aerosol modes of defined dry size distribution and composition can be initialised, and the simulated air parcel rises at a prescribed updraft velocity. ACPIM

uses bin microphysics and thermodynamics to evaluate the rate of condensation of water in each size bin numerically using the diffusional drop growth equation and Köhler theory (see Topping et al., 2013), so does not rely on bulk parametrizations of cloud drop activation. The number of particles in each aerosol mode growing above the critical diameter is then evaluated to determine the CDNC.

ACPIM was initialised starting at 95% humidity, 296 K, and 960 hPa, which are typical of conditions

just below cloud base over Togo. Aerosol size distributions from SMPS and relative compositions from AMS were averaged over regions at various distances offshore or inland over SWA, and these were used to initialise the model for several runs at different updraft velocities. The hygroscopicity parameter ($\kappa$) values of inorganic and organic aerosols were assumed to be the same as ammonium sulfate and fulvic acid, which is chemically similar to highly aged organic aerosol (Jimenez et al., 2009). By using fixed, representative

thermodynamic starting conditions, varying the updraft velocities an aerosol number, size, and composition, allows us to determine the relative sensitivities of CDNC to both variables.

Our simple modelling scheme does not include factors such as entrainment, collision-coalescence, or an investigation into the impacts of aerosol mixing state, which can have a large effect on calculated CCN (Ren et al., 2018). We have also only considered a limited number of aerosol species, and other components

such as black carbon, sea salt, and mineral dust will have been present. This is not intended to be an exact simulation, but a sensitivity analysis using physically reasonable approximations. A detailed study of aerosol activation is beyond the scope of this analysis and is non-trivial (e.g. Sanchez et al., 2017). Additionally, we have no measurements with which to constrain factors such as the size-dependence of chemical composition and mixing state, which strongly affect the accuracy of CCN closure calculations (Moore et al., 2013), so

such an investigation would be purely speculative.


## 3 Results

### 3.1 Diurnal cycle of low-level cloud cover

Figure 2 shows the average diurnal cycle of LLC fraction over Togo and Benin, where the aircraft operated, plotted versus distance North of the coast and approximate latitude. LLC peaked around 10:00, and fell to

a minimum around 18:00. The initial surface warming in the few hours after sunrise causes the nocturnal clouds to rise and thicken, before eventually breaking up (van der Linden et al., 2015; Kalthoff et al., 2018). Cloud cover was generally greater inland than offshore, except for a period in the late afternoon / early evening. A region of more extensive cloud cover was seen developing overnight, around 50 – 150 km inland. This is the region the maritime inflow reaches on a typical evening, though it is interesting that cloud cover

did not increase overnight nearer the coast. The highest LLC fraction was around 250 km inland, in a region that began to develop overnight but became particularly pronounced during daylight hours. A further region of higher cloud cover was seen just inland of the coast. These are most likely sea breeze clouds developing from mid-morning to the early afternoon, and moving up to ~50 km inland before dissipating. Lower levels of cloud cover (average 0.24) were seen over all inland areas between 16:00 and 00:00. By taking the mean

over several weeks we lose the extremes of these values- on some evenings cloud cover was zero and on some mornings it reached 100%. This averaging allows us to assess which features are statistically robust, and minimises transient features in the cloud field. Figure 2 bears a striking similarity to similar diagrams of boundary layer temperature and relative humidity presented by Deetz et al. (2018a), suggesting transport of cool, humid air inland by the maritime inflow is a key factor in determining LLC cover.

### 3.2 Regional variation in low-level cloud cover

Figure 3 shows the orography of West Africa, and a map of LLC fraction at its 10:00 UTC peak. The LLC fraction decreases dramatically in the drier regions north of ~10 – 11°N. South of this latitude, the largest LLC fractions were seen on the upwind side of slopes, and the lowest on the leeward sides, for a southwesterly monsoon flow. Additionally, a patch of lower LLC fraction is seen just offshore of Ghana,

Togo and Benin, which is related to the colder waters there due to the coastal upwelling system (see Flamant et al. (2018a)). The features in our mean LLC fraction over the region are broadly similar to those presented in multi-year satellite observations by van der Linden et al. (2015), meaning the measurement period is





likely to be broadly representative of a typical monsoon season. The absolute values we present are more in line with the synoptic observations than the satellite measurements in the previous work, as the LLC fraction product here takes into account times when LLC would not be visible due to higher cloud obscuring the view. Figure 3 also shows the major population centres in the region. Despite the presence of urban

emissions, anthropogenic aerosols had no clear effect on cloud cover downwind of the major cities, and this was the case in plots similar to Fig. 3 at all times of day.

### 3.3 Regional variation in cloud microphysics

To investigate the effect of local emissions on cloud microphysical properties, we collated cloud measurements below 1 km in altitude from all three aircraft and compared them with distance to the coast (and the

population centres nearby) on a north/south axis, as shown in Fig. 4a. Figure 4b shows corresponding normalised histograms in the different regions. The heterogeneous nature of the dataset means that some areas and some days have large numbers of data points recorded, while others have relatively few. In particular, cloud measurements over the sea were sparse compared to inland. To give an indication of how statistically representative our data are, the number of data points and number of individual days the data are from, are

also listed in Fig. 4a.

Offshore, we split the data up into two bins. The aim of the offshore analysis is to consider clouds and pollution conditions representative of the air upwind of the DACCIWA region, that is then brought inshore by the wind. Close to the coast, several factors may cause terrestrial pollution to be transported to an area offshore. Firstly, some areas of the coast are more prominent than others, for example the area located just

offshore near Lomé may be downwind of Accra if the wind was blowing from a west-south-westerly direction. Secondly, sea breeze circulations may act to recirculate pollution (Flamant et al., 2018a). Additionally, a shipping corridor near to the coast presents a local source of CO and aerosol. In Fig. 4a there is no apparent effect of any pollution near to the coast, but plumes of CO and aerosol were observed up to 20 km offshore that would affect the analysis in Sections 3.6 and 4. For the remainder of this analysis, when con-

sidering "'offshore"' data, we therefore only consider measurement made at least 20 km south of the coast, to unambiguously remove any possibility of local terrestrial emissions affecting our offshore measurements.

In the offshore bin, there were CDNC measurements from 5 individual days, and the distribution of measured CDNC in Fig. 4b appears bimodal, with a main mode centred around $100 - 200 \text{ cm}^{-3}$, and a smaller





mode centred around 500 cm$^{-3}$. Looking at Fig. 4a, this bimodality is the result of sampling different populations of clouds. This may be due to variable pollution conditions offshore, or some other factor. Bennartz (2007) studied CDNC in the global marine boundary layers, and found average CDNC in the pristine south Atlantic was 67 cm$^{-3}$, reported in ambient temperature and pressure. For stratocumulus topping the south

Atlantic marine boundary layer at 800 hPa and 280 K, this number is roughly equivalent to 87 cm$^{-3}$ when corrected to STP, which lies around the 10th percentile of the CDNC measured in offshore clouds in the DACCIWA region, meaning the offshore measurements presented here were representative of moderately polluted clouds.

Moving inland, several differences are apparent. The inland CDNC distributions in Fig. 4b are approxi-

mately Gaussian, centred around ~400 cm$^{-3}$, but with a long tail extending up to over 1000 cm$^{-3}$, which was measured most frequently near the urban centres near the coast, and diminished further from the coast. Some clouds were observed with CDNC over 1500 cm$^{-3}$. This long tail on the distribution is due to the effect of thick urban plumes, but it only represents a relatively small fraction of the inland data. The large majority of CDNC measurements inland were in the range 200 – 840 cm$^{-3}$ (10th and 90th percentiles), and

medians were 470 cm$^{-3}$ up to 50 km from the coast, and ~430 cm$^{-3}$ further inland. It is possible that some of this small difference in the medians may be due to measurement of more convective sea breeze clouds near the coast, but it is relatively minor overall.

To investigate the effect of local pollution on CDNC, Fig. 4c shows the distribution of CO concentrations measured in inland clouds. The inland CO data show a mode with mean and standard deviation of

141 ±17 ppbv, but the distribution is asymmetric, with a tail extending to higher values above around 160 ppbv. We therefore used this value of 160 ppbv as a threshold to distinguish the thickest pollution plumes. For comparison, the mean and standard deviation of in-cloud offshore CO concentrations were 143 ± 11 ppbv. This offshore value, and similarly the minimum inland CO concentrations of ~100 ppbv, is significantly enhanced compared to CO concentrations in the unpolluted south Atlantic, which reach lows

of 60 ppbv in the absence of transported biomass burning smoke (Zuidema et al., 2018). This contrast highlights the ubiquity of transported biomass burning smoke affecting the DACCIWA region (Haslett et al., submitted, 2019).

Using the 160 ppbv CO threshold to stratify the CDNC highlights the effect of pollution plumes, which we show in Fig. 4d. Using only the inland data here removes any possible bias from differing conditions (for


example dynamics) over the sea. The data in Fig. 4d show enhanced CDNC in the polluted plumes, with the entire distribution shifted around 1.5 times higher. However, these highly polluted plumes represent less than 10% of the inland data measured over the region.

There appeared to be a difference between the offshore and inland clouds that was not related to differing
pollution. Inland clouds with CO < 160 ppbv (i.e. similar to levels found offshore) had a median CDNC of $430\,\mathrm{cm}^{-3}$, compared to $265\,\mathrm{cm}^{-3}$ offshore. However, Fig. 4b shows some offshore clouds that had CDNC and CO comparable to those found inland. With the limited offshore data available we are unable to fully quantify whether there was any systematic difference, which might be expected due to different dynamical conditions.

## 3.4   Vertical profile of cloud microphysics

Figure 5 shows average vertical profiles of $N_{250}$, cloud drop number concentration, and cloud effective radius, measured at different distances north or south of the coast. The aerosol measurement here is only of particles larger than 250 nm, so it is representative of the variability of the accumulation mode, though not the total number. It does not show any variability related to smaller particles. Inland, accumulation mode aerosol
concentrations were highest closer to the ground, and decreased with altitude up to around 2 – 2.5 km. There was relatively little meridional variation inland, but aerosol concentrations were enhanced by around ~40% near the ground compared to offshore concentrations. This enhancement decreased with altitude, as concentrations offshore were fairly constant up to 2.5 km, and above 2.5 km offshore concentrations were higher than those inland. In the boundary layer, the southwesterly winds brought offshore pollution inland,
and local city emissions would be added to the pollution transported from offshore, but above the inversion at boundary layer top, the wind shifted to a regime dominated by the African easterly jet and this linkage was broken. Above the boundary layer, distinct biomass burning plumes were encountered more often, which caused the larger variability in $N_{250}$ in the free troposphere.

The CDNC profile inland (Fig. 5b) shows a decrease with altitude, with a similar profile to $N_{250}$, but
the relative decrease was lower in magnitude. Inland there was high variability within all regions, but little difference between the average values of CDNC, though >100 km inland concentrations were ~10% lower than closer to the coast. Offshore concentrations at almost all levels were significantly lower than those inland, and the difference was larger than the differences in aerosol and did correlate with differences in



the aerosol vertical profiles. However, the majority of offshore bins in Fig. 5b show data from one or two individual days, so may not be representative of offshore clouds in general. If the difference is representative, it does not appear to be directly related to differences in accumulation mode aerosols.

Figure 5c shows the corresponding cloud effective radius profile. Again, there was relatively little differ-

ence inland, particularly in the boundary layer, and on the whole $R_{\mathrm{Eff}}$ increased with altitude. In the free troposphere there was more deviation, and did not always increase with altitude. This due to the structure of clouds over SWA; multiple cloud layers with distinct and separate bases were often present, meaning clouds at higher altitudes were not necessarily deeper than those below. Between 0.5 and 1.5 km, the clouds offshore tended to have higher $R_{\mathrm{Eff}}$ than those measured inland at the same altitude, which is consitent with

the lower CDNC measured in this altitude range. However, throughout the rest of the lower troposphere, including near the surface, the clouds measured offshore had similar $R_{\mathrm{Eff}}$ to those measured inland at the same altitude, despite generally lower CDNC. This implies that offshore clouds had lower liquid water contents than inland clouds at the same altitude, suggesting that clouds offshore may have been influenced by a different thermodynamic structure, such as different temperature or humidity profiles causing different cloud

base heights.

Overall, relatively little systematic spatial variability is seen inland in aerosol or cloud parameters. There are differences in aerosol and CDNC between inland and offshore clouds, but not in $R_{\mathrm{Eff}}$, meaning the differences cannot be solely due to increased aerosol activation inland. For the remainder of this analysis we will consider only clouds measured in the lowest kilometre of the atmosphere, as this covers the majority

of the cloud measurements made in the morning, and to minimise the influence of the African easterly jet. Although the properties of individual clouds were variable, the average inland clouds below 1 km were fairly homogeneous regardless of the distance inland, meaning we can consider all inland clouds together and develop better statistics of the day-to-day variability and diurnal cycle of cloud properties.

### 3.5   Day-to-day variability

In the previous sections, we demonstrated that the ubiquitous background aerosol reaching SWA caused all clouds to be fairly polluted. Figure 6 shows the day-to-day variability in both the offshore and inland aerosol concentrations, as well as the CDNC. For the accumulation mode aerosol, the concentrations offshore were 50 – 90% of those inland, and the offshore and inland concentrations were well correlated. The mean and



standard deviations in daily $N_{250}$ are $170\pm50\ \mathrm{cm}^{-3}$ offshore and $230\pm50\ \mathrm{cm}^{-3}$ inland, meaning the average variabilities were similar. Together, this suggests that aerosol imported from offshore comprised the majority of accumulation mode aerosol inland, and also strongly influenced its day-to-day variability.

The day-to-day variability in the aerosol was not strongly represented in the CDNC measurements in
Fig. 6b. There was a degree of correlation in the median measurements after the 5th July, but there was no correlation in the first half of the campaign. Comparing the inland CDNC to offshore, on four days the median values inland were 85 – 175% higher than those measured offshore, but 25% lower on one day. These differences were not correlated with differences in the offshore and inland aerosol. As was stated in Sect. 3.3, we do not have sufficient measurements to investigate or explain these differences in detail, or to
assess if they are representative. On any individual day, the CDNC measurements were much more variable than the aerosol concentrations, suggesting highly localised factors such as entrainment or dynamics may have had an influence on the CDNC. Overall, the day-to-day variability in the inland CDNC and $N_{250}$ were both ~20%, and there was a degree of correlation for some of the project, suggesting aerosol concentrations may have had some effect on daily CDNC. In the next Section, we discuss diurnal variation, to investigate
these factors in more detail.

## 3.6   Diurnal variability of aerosols, clouds, and dynamics

The variation in inland CDNC at different times of day is shown in Fig. 7a. As some of the measurements were taken over a wide range of longitude, here we used the local solar time (LST) instead of UTC. A trend is apparent in the data, with CDNC increasing inland throughout the day and peaking around 15:00 LST.
The inland CDNC measurements were 45 – 60% higher in the afternoon than before 08:00. The median of all CDNC measured inland in the morning was $430\ \mathrm{cm}^{-3}$, and $540\ \mathrm{cm}^{-3}$ in the afternoon. We draw this distinction because cloud cover peaked in the late morning, before decreasing in the afternoons as the clouds broke up. The 90th percentiles of CDNC in Fig. 7a exhibited a more dramatic increase in the afternoon, indicating an greater cloud drop concentrations in the more convective broken clouds.
CDNC is determined by the supersaturation in an updraft, and the concentration of aerosols that activate at that supersaturation. Figures 7b & c show the diurnal variations in $N_{250}$ and in-cloud vertical velocity. The accumulation mode aerosol showed no strong diurnal variation, varying only $\pm10\%$ during the day. In comparison, the distributions of vertical velocity showed a shift towards stronger updrafts in the afternoon.





In any stable cloud layer the median vertical velocity is zero, and this was the case for all times of day
shown in Fig. 7c other than the few cloud measurements made after 16:00 LST. As the median value of
vertical velocity in a stable cloud is zero, the 75th percentile can be considered as a representative updraft,
and is useful to investigate changes in updrafts that are present at cloud base, where the majority of aerosol
activation occurs.

The distribution of vertical velocity was fairly stable throughout the morning, with 75th percentiles around
0.45 $\mathrm{m\,s^{-1}}$, and 90th percentiles around 1 $\mathrm{m\,s^{-1}}$. In the afternoon, the clouds became more convective,
with 75th percentiles of vertical velocity of 0.5 – 1.6 $\mathrm{m\,s^{-1}}$ and 90th percentiles of 1.1 – 1.7 $\mathrm{m\,s^{-1}}$. The
stronger updrafts in the afternoon suggest that dynamics, rather than varying aerosol concentrations, were
responsible for the increase in CDNC seen in the afternoons over SWA. In the next Section we use parcel
model simulations to quantify the effect of these factors on CDNC.

## 4   Parcel model simulations

To further investigate the relative sensitivities of CDNC to aerosol properties and updraft velocities, we con-
ducted a series of parcel model simulations using ACPIM, described in Sect. 2.1.2. We initialised the model
with representative thermodynamic conditions and varied the updraft velocity and aerosol size distribution
and composition based on the measurements made in the field campaign.

### 4.1   Aerosol size and composition

The average measured aerosol size distributions are shown in Fig. 8a – c. Here, the data coverage was
more limited, and the averaging time of the instrument was 90 s, so a simple mean of all the data at each
latitude relative to the coast was used, averaging out any diurnal or day-to-day variation. The measured
offshore average aerosol size distribution and accumulation mode composition are shown in Fig. 8a & d.
Two lognormal fits were used to approximate this size distribution for use in ACPIM. The measured inland
average aerosol size distribution and accumulation mode composition are shown in Fig. 8c & f. The offshore
size distribution was subtracted from the inland, to give an estimate of the average contribution of dispersed
local emissions to regionally-averaged inland aerosol over SWA. Two lognormal fits were then used to
approximate the average local emissions, and the sum of the four fitted modes makes up the measured size





distribution inland. The lognormal fit parameters are listed in Table 2. The utility of this approach is it allows us to scale the transported background aerosol (i.e. the offshore fits) and the local aerosol separately.

The measured size distributions in Fig. 8a & c, and the measured compositions in Fig. 8d & f, are almost identical to similar estimates presented by Haslett et al. (submitted, 2019). The difference in our approach
when highlighting the impact of local emissions is that we have subtracted the transported background to isolate the effect of dispersed local emissions on regional inland aerosol, whereas Haslett et al. (submitted, 2019) measured aerosol in the thickest urban plumes, but still on top of the transported background.

The offshore aerosol size and composition are consistent with being a mixture of aged biomass burning (Rissler et al., 2006; Capes et al., 2008; Sakamoto et al., 2014) and Atlantic marine aerosol (Zorn et al.,
2008; Taylor et al., 2016). The two modes of local aerosol were an Aitken mode containing the majority of the particle number concentration, and a smaller accumulation mode. We are unable to determine the specific source of each mode, but local emissions are largely from domestic wood burning and transport, and when combined these sources are capable of producing aerosols with this size distribution and composition (Maricq et al., 2000; Capes et al., 2008; Vakkari et al., 2014). More informative studies of aerosol size dis-
tribution in African cities appear to be absent from literature. The local Aitken mode, which accounts for the majority of particle number inland, contains particles that were too small to make a significant contribution to CCN in stratocumulus clouds, and the majority of the aerosol that are large enough to activate into cloud drops were still the result of long-range transport.

## 4.2  Model results

We used the aerosol size distributions and composition shown in Fig. 8 and Table 2 as input for ACPIM simulations at varying updraft velocities. The resultant simulated CDNCs are shown in Fig. 9a. For weak updrafts the different input aerosols makes little difference to the CDNC values, as particle activation starts from the largest sizes, where the inland and offshore size distributions were most similar. The difference between the two then increases with updraft velocity. For the representative morning updraft of $0.45\,\mathrm{m\,s^{-1}}$,
calculated as the 75th percentile of vertical velocity, the calculated enhancement in inland CDNC due to local aerosols is 44%. This enhancement is larger for stronger updrafts; for the representative afternoon updraft of $0.78\,\mathrm{m\,s^{-1}}$, it rises to 65%. In polluted clouds with relatively gentle convection, CDNC is more sensitive to changes in updraft that in aerosol (Reutter et al., 2009). However, Fig. 9a suggests the clouds over SWA are



only in this regime for the lowest updrafts, and for the average updrafts the main sensitivity of CDNC is not so clear-cut.

A large fraction of the day-to-day variability expected in the mean inland CDNC, shown by the shadings in Fig. 9, derives from day-to-day variability in the offshore aerosol brought inland. Using these average

updrafts, the modelled CDNC is 31% higher in the afternoon than in the morning, which compares well to a 26% increase in the measured CDNC shown in Fig. 7.

Fig. 9b shows the effect of varying the local emissions on the calculated CDNCs, using these average morning and afternoon updrafts. For comparison, the top axis in Fig. 9b shows the number concentration of aerosols larger than 125 nm ($N_{125}$) in the input size distributions, as a way of comparing the effects that

scaling local emissions have on aerosol concentrations (i.e. transported background plus local emissions). The diameter of 125 nm was chosen as $N_{125}$ is roughly comparable to the modelled values of CDNC in Fig. 9b, and is therefore an approximation of the modelled critical diameter, although the actual critical diameter will have been different in each model run, and varies as the aerosol size distributions and number concentrations change.

An alternative way of looking at the impact of transported aerosols inland is to say that if local emissions were doubled, $N_{125}$ would only be 38% higher, and CDNC would only increase by 17% in the morning and 22% in the afternoon. If local emissions were halved, $N_{125}$ would be reduced by 19%, and CDNC would decrease by 13% in the morning and 16% in the afternoon. If local emissions were removed entirely, $N_{125}$ would decrease by 38%, and CDNC would drop by 31% in the morning and 39% in the afternoon.

Large changes in local emissions cause smaller changes in CDNC because the high aerosol background means they form only a fraction of the total aerosol. Additionally, the local emissions are smaller and less hygroscopic than the background aerosol, so they are less likely to activate. The effect of varying local aerosol emissions does not produce a linear relationship between CDNC and total aerosol concentrations. In particular, the effect of increasing total aerosol concentrations has less of an impact on CDNC than

decreasing aerosol. The exact comparability between the aerosol metric and CDNC will depend on the size assumed as representative of the critical diameter (125 nm in this comparison, but would vary as the aerosol changes), but the nonlinearity of the relationship between aerosol and CDNC is independent of this choice of size.

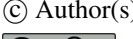



The values of CDNC produced by ACPIM are higher than the median values shown in the ambient measurements. We have acknowledged the limitations of our simplistic approach above, such as the assumptions regarding aerosol mixing state and the lack of entrainment in the model. Moore et al. (2013) summarised 36 previous studies of CCN closure, and similarly found that CCN concentrations were generally overpre-
dicted, other than when using size-resolved measurements aerosol composition. Additionally, a previous comparison by Simpson et al. (2014) showed that ACPIM calculated higher CDNC compared to simpler parameterizations when using multiple aerosol modes. The values of modelled CDNC are between the 75th and 90th percentiles of measured CDNC shown in Fig. 7a. It may be that entrainment ubiquitously affected clouds sufficiently to reduce the median CDNC, as the variability in CDNC on a particular day was much
larger than the variability in accumulation mode aerosol. We have used a threshold for in-cloud data of $0.1\,\mathrm{g\,m^{-3}}$ and found a campaign median inland CDNC of $450\,\mathrm{cm^{-3}}$. Using a threshold of $1\,\mathrm{g\,m^{-3}}$ gives a value of $720\,\mathrm{cm^{-3}}$. This suggests that entrainment is likely to have affected CDNC, otherwise CDNC would not vary with LWC. Convective clouds may have an adiabatic core that is relatively unaffected by entrainment (Heymsfield et al., 1978), but this is less likely to be the case in layer clouds that form the majority of
our dataset.

## 5   Discussion

Most stratocumulus clouds are over the oceans, and most previous studies of cloud-aerosol interactions in stratocumulus clouds have focused on clouds in this setting. Allen et al. (2011) observed CDNC in the southeast Pacific decreasing from $300\,\mathrm{cm^{-3}}$ near the coast to ~$100\,\mathrm{cm^{-3}}$ $150\,\mathrm{km}$ downwind offshore. Their
situation is simpler to visualise as the anthropogenic source regions are well defined, the ocean surface is flat, and there is no strong diurnal cycle in the marine boundary layer. In SWA, the situation is made more complex by the combination of local sources and the variability of the dynamics, both in terms of the diurnal cycle and the difference between offshore and inland.

In very remote regions of the south Atlantic, like in the southeast Pacific, clouds are generally clean, with
average cloud drop number concentration estimated at $67\,\mathrm{cm^{-3}}$ (or $87\,\mathrm{cm^{-3}}$ when corrected to STP) (Bennartz, 2007), and effective radius may be ~$15 - 20\,\mathrm{\mu m}$ for clouds up to $1.5\,\mathrm{km}$ in height (Painemal et al., 2014). In this regime, shortwave cloud reflectance is strongly susceptible to change in aerosol concentra-





tion. In DACCIWA, the limited measurements of offshore CDNC were split into clouds with around $100 - 200\,\mathrm{cm}^{-3}$ and those with around $200 - 840\,\mathrm{cm}^{-3}$. None of these measurements were representative of clean clouds, as biomass burning smoke from fires in western Central Africa was ubiquitous off the coast of SWA. This transported pollution was brought inland at low-levels by southwesterly monsoon winds, and therefore

also affected the properties of low-level cloud over the African continent.

The polluted background has three main consequences for the microphysics of low-level clouds. Firstly, it means local emissions are a lower fraction of the total aerosol loading and of CCN concentrations at any given supersaturation, limiting the impact of local emissions on CDNC. Secondly, with the baseline CDNC at already-polluted values, it puts the clouds in a regime where CDNC is less sensitive to increases in

aerosol concentrations than decreases. Finally, cloud albedo is not overly sensitive to changes in CDNC that do occur when the baseline is so high (Twomey, 1991), so this further limits the effect of increasing local aerosol emissions on cloud albedo.

The day-to-day and diurnal variability are extra factors that complicate our analysis, but they do not detract from this overall picture. Similar results were found by Cecchini et al. (2017) in the polluted regions of

southern Amazonia, affected by widespread biomass burning, in contrast to the cleaner coastal and forested areas. The modelling results in Fig. 9 show that the increase in CDNC in the afternoons is broadly that expected from the increased updraft speeds. With the limited data available, it is difficult to assess whether or not there is a systematic difference between CDNC or dynamics offshore versus inland. While Fig. 4d shows an increase in CDNC in pollution plumes, the CO distribution inland did not show any large systematic

increase in pollution to the offshore mean and standard deviation of $141 \pm 7\,\mathrm{ppbv}$, meaning the bulk of the measurements shown in Fig. 4b were made in similar pollution conditions. It may be that our poor statistics of offshore data present a misleading distribution in Fig. 4b, or that there was a systematic and ubiquitous increase in CDNC due to some other factor such as stronger updrafts inland. Most of our offshore measurements were made over the patch of cooler water with lower cloud cover shown in Fig. 3, which

may have affected the offshore measurements. Regardless, our evidence is not consistent with a ubiquitous increase in CDNC inland as a result of a large increase in aerosols due to local anthropogenic pollution.

Across the whole of SWA, anthropogenic emissions are dominated by Nigeria, which contains over half the region's population (United Nations, 2017). The Niger delta is also a large source of industrial pollution. In the boundary layer, Nigeria was downwind of our in situ measurements, so we have no measurements of



Nigerian pollution. Under such conditions, Flamant et al. (2018a) and Deroubaix et al. (2018) have shown that Lagos did not influence on the air quality to the west (i.e. Benin and Togo). It is possible that local emissions have more of an effect on CDNC in Nigeria, but the baseline CDNC is still likely to have been high enough that this will have a limited effect on cloud albedo. Based on the satelite data there was no obvious effect on LLC fraction downwind of Lagos and the Niger delta, suggesting geographical variation in cloud formation and lifetime were still largely controlled by dynamics, orography, and the transport of cool, humid air inland by the maritime inflow.

Over the next few decades, the populations of coastal cities such as Accra, Lomé and Cotonou are expected to rise dramatically, as are their respective emissions. While this may mean there is more of an effect from local pollution on the cloud field, our results suggest the dominating factors for cloud radiative properties during the summer West African monsoon are likely to remain the transport of biomass burning aerosol from the southern hemisphere. Indeed, as the population of western Central Africa is also expected to rise at a similar rate to SWA (United Nations, 2017), increasing rates of agricultural burning may cause this background to rise. In this study we have focused on cloud-aerosol interactions leading to the indirect effects, and have not attempted to investigate possible changes to the cloud field by the semi-direct effect or direct effect via changing dynamics, as suggested by Deetz et al. (2018b) and Kniffka et al. (2018). It is vital that future studies examining these effects take into account the transport of biomass burning emissions to SWA from the southern hemisphere.

## 6 Conclusions

This study has assessed factors affecting properties of low-level clouds over southern West Africa during the West African monsoon. Satellite observations of low-level cloud cover suggested regional variation in the cloud field was determined largely by the local orography and transport of cool, humid air inland by the maritime inflow, and no obvious impact of anthropogenic aerosols was observed in the large-scale cloud field downwind of major population centres. An assessment of cloud drop number concentration based on aircraft data acquired during the DACCIWA field campaign as well as parcel modelling showed that local emissions had a limited effect on CDNC on the regional scale, as biomass burning pollution transported from the southern hemisphere dominated regional aerosol concentrations (e.g. Haslett et al., submitted, 2019). This



transported pollution also caused a high baseline CDNC inland, putting the clouds in a regime where they had a limited susceptibility to any further increases in aerosol, and also minimising the impact any enhancements in CDNC are likely to have on cloud radiative effects and precipitation.

We investigated statistics of CDNC on different days, at different times of day, and at different distances
downwind of the coast (and the populated regions nearby), to assess the causes of the observed variability. Overall, the large majority of CDNC fell into the range $200 – 840 \, \mathrm{cm}^{-3}$, and relatively little meridional variation was seen inland. The day-to-day variability of CDNC showed some correlation with measured aerosol concentrations, but this was not seen across the whole of the campaign. A diurnal cycle was observed, and CDNC increased by 45 – 60% as clouds began to break up and become more convective in the afternoons
compared to measurements made in the early morning. A systematic increase in CDNC was observed in CO plumes near the coastal cities, reaching over $1500 \, \mathrm{cm}^{-3}$ in some cases. However, these plumes represented only a small fraction of the data measured across the region, and are unlikely to be the major factor determining CDNC inland. Offshore, the limited measurements of CDNC were split into clouds with around $100 – 200 \, \mathrm{cm}^{-3}$ and those with around $300 – 700 \, \mathrm{cm}^{-3}$. However, relatively few cloud penetrations were made
offshore, so it is difficult to assess whether any differences between offshore and inland clouds are statistically representative. In no region, and at no time of day, were clouds observed that were representative of those found in pristine environments, and our measurements were all affected by the ubiquitous biomass burning pollution transported to the region from offshore.

We performed a sensitivity analysis with a parcel model, which was initialised with representative aerosol
size distribution and composition, and thermodynamic conditions based on the ambient measurements. We divided up the aerosol into different modes representing either transported pollution or local emissions, and scaled the local emissions to see the effect on CDNC. Doubling local emissions increased the calculated CDNC by 17 – 22%, whereas it was reduced by 13 – 16% if local emissions were halved, or by 31 – 39% if they were removed altogether. The high aerosol background from transported smoke means local emissions
only make up a fraction of total aerosol. Our results suggest that increasing local emissions over the next few decades will therefore have a limited impact on CDNC.

Compared to clean conditions, this polluted regime means local emissions have a more limited effect on aerosol concentrations, enhanced aerosols have a more limited effect on CDNC, and enhanced CDNC has a more limited effect on cloud radiative properties. As the southern hemisphere biomass burning sea-



son and West African monsoon coincide every year, and the cloud field during DACCIWA was similar to a previous multi-year assessment by van der Linden et al. (2015), it is reasonable to assume that our observations are typical of clouds over SWA at this time of year. As the population of SWA grows in the coming decades, the transported biomass burning pollution will have a dampening effect on the impacts that grow-
ing aerosol emissions have on low-level clouds via cloud-aerosol interactions. Future studies investigating other mechanisms by which aerosols may affect the cloud field, such as by changing atmospheric dynamics and thermodynamics, must take into account the transport of biomass burning smoke from the southern hemisphere to SWA.

*Code and data availability.* The DACCIWA field measurements are available on the DACCIWA SEDOO database
(http://baobab.sedoo.fr/DACCIWA/). SEVIRI optimal cloud analysis data and associated product guides are available to download from EUMETSAT (https://www.eumetsat.int/website/home/Data/Products/Atmosphere/index.html). ACPIM is available to download from The University of Manchester (https://personalpages.manchester.ac.uk/staff/paul.connolly/research/acpim01.html).

## Appendix A:  Low-level cloud flag derivation

The SEVIRI optimal cloud analysis product is based on the scheme described by Watts et al. (2011). Cloud top pressure (CTP), cloud optical thickness (COT), and cloud-top effective radius, and their respective uncertainties, are reported every 15 minutes with a spatial resolution of around 3 km in the DACCIWA region. Here we utilise the CTP for the upper and second cloud layers ($CTP_1$ and $CTP_2$), and the COT of the upper layer ($COT_1$).
Our scheme for determining LLC cover is shown in Fig A1. Pixels are defined as LLC if either $CTP_1$ or $CTP_2$ are statistically significantly above a threshold pressure ($CTP_T$). Where either pressure is within the uncertainties of this threshold, the pixel is defined as "'borderline"' LLC. Additionally, where the top layer of cloud is not at low level, a threshold COT ($COT_T$) is used to account for the fact that this thick top layer makes it difficult to detect any other cloud layers below. This extra factor may be responsible for the better
agreement between our scheme and ground measurements compared to others (e.g. van der Linden et al., 2015). We chose a value of $CTP_T$ of 680 hPa (~3.5 km) because the OCA algorithm can sometimes place



a cloud on the wrong side of an inversion. 680 hPa is near a minimum in relative humidity (Kalthoff et al., 2018), at a level above the inversion at boundary layer top but below any mid-level clouds.

A comparison to LLC fraction derived from ground-based ceilometers in Kumasi and Savé is shown in Fig. A2. The ceilometers made ground-up measurements of cloud base, compared to top-down satellite mea-

5   surements of cloud top, which may cause a small uncertainty in the comparison that is difficult to quantify. Surprisingly, the best agreement was generated by ignoring the per-pixel uncertainty in the SEVIRI data (i.e. assuming all values of CTP were absolutely accurate). This comparison showed agreement in the diurnal cycle and absolute values within 10% LLC fraction, and similar levels of variability (i.e. the standard deviations). The comparison was better during the day than at night, in agreement with a similar observations by

10   van der Linden et al. (2015), though using a different LLC scheme.





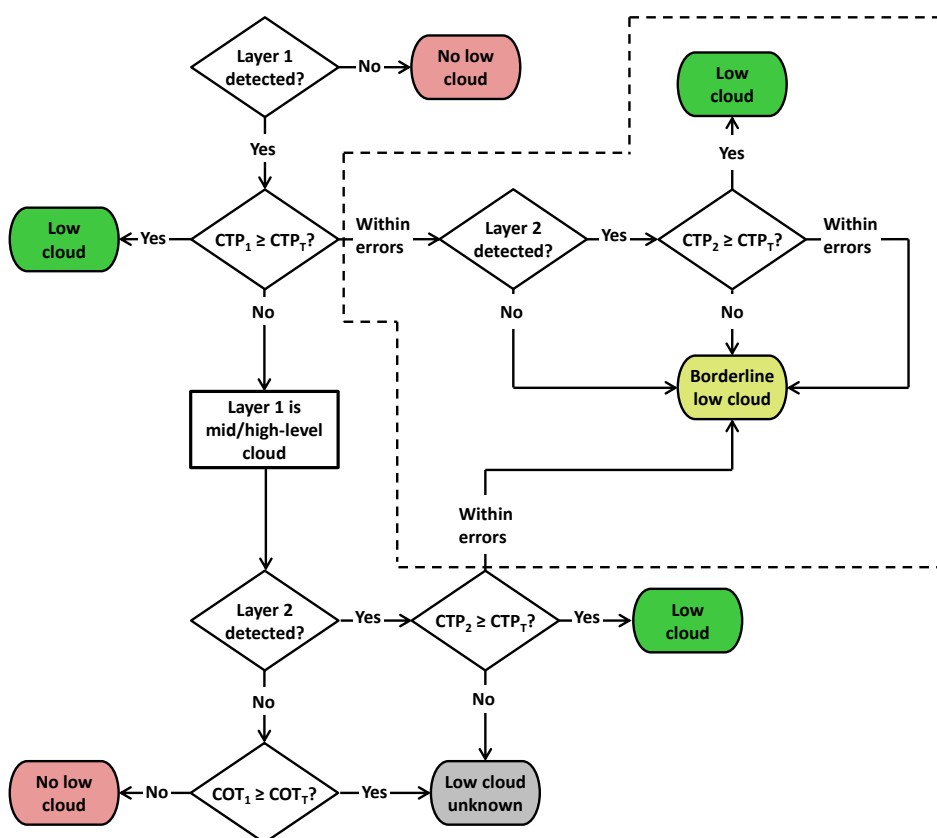

**Figure A1.** Schematic describing the derivation of the LLC flag from SEVIRI OCA data. We used values of $CTP_T = 680$ hPa and $\log_{10}(COT_T) = 0$. The dashed lines separate off the section of the scheme that is only necessary where the per-pixel uncertainty is considered.



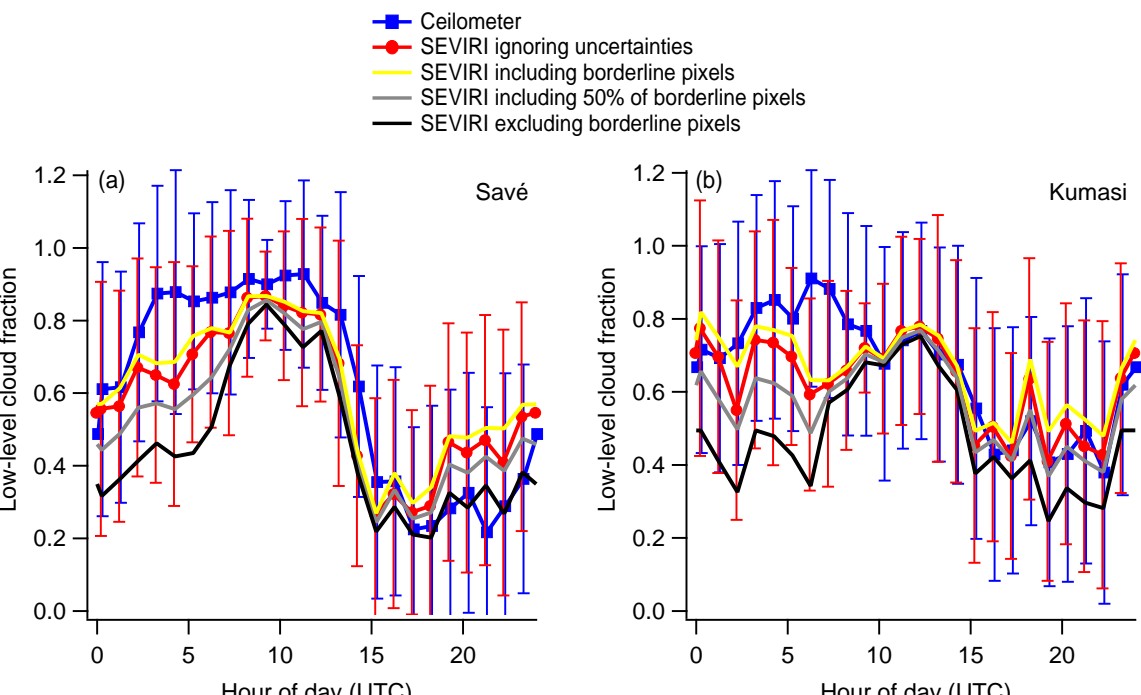

**Figure A2.** Comparison of LLC fraction derived from satellite and ground-based ceilometer measurements from Savé (a), and Kumasi (b). The satellite measurements were calculated from a $4 \times 4$-pixel grid centred on the ground site.





*Author contributions.* JWT prepared the manuscript and performed the bulk of data analysis, with input from all coauthors. JWT, SLH, KB, MF, IC, JD, CV, VH, DS, RD, JB, AS, TB, CD, PR, CF, JDL, ARV, VC, and PK carried out the airborne measurements. JWT, SH, VH, DS, RD, JB, TB, JDL, ARV, and VC processed the aircraft data. BB carried ground measurements and provided the Kumasi ceilometer data. TC, PJC, and PGH advised on the data analysis. PK, HC, and CF are lead PIs who led the funding application and directed the research.

*Competing interests.* The authors declare no competing interests.

*Acknowledgements.* The research leading to these results has received funding from the European Union 7th Framework Programme (FP7/2007-2013) under Grant Agreement no. 603502 (EU project DACCIWA: Dynamics-aerosol-chemistry-cloud interactions in West Africa). Sophie Haslett was also supported by a PhD studentship from the UK Natural Environment Research Council. Cristiane Voigt and Valerian Hahn acknowledge funding by the Helmholtz Society under contract W2/W3-60 and by DFG-SPP2115 PROM under contract VO1504/5-1. We thank British Antarctic Survey (BAS, operator of the Twin Otter), the Service des Avions Français Instrumentés pour la Recherche en Environnement (SAFIRE, a joint entity of CNRS, Météo-France, and CNES and operator of the ATR 42), and the Deutsches Zentrum für Luft- und Raumfahrt (operator of the Falcon 20) for their support during the aircraft campaign. PJC acknowledges funding from the European Union's Seventh Framework Programme (FP7/2007-2013) under grant agreement no. 603445 (BACCHUS). We thank Norbert Kalthoff and Bianca Adler for providing the Savé ceilometer data, and Lynn Hazan for the ATR CO data.



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



**Table 1.** List of cities in Fig. 3.

| Code | City name | Latitude (°) | Longitude (°) |
|------|-----------|-------------:|--------------:|
| AB | Aba | 5.12 | 7.37 |
| AC | Accra | 5.55 | -0.20 |
| AJ | Abidjan | 5.32 | -4.03 |
| BE | Benin | 6.33 | 5.62 |
| CO | Cotonou | 6.37 | 2.43 |
| IB | Ibadan | 7.40 | 3.92 |
| IL | Ilorin | 8.50 | 4.55 |
| JO | Jos | 9.93 | 8.88 |
| KA | Kano | 12.00 | 8.52 |
| KD | Kaduna | 10.52 | 7.44 |
| KU | Kumasi | 6.67 | -1.62 |
| LA | Lagos | 6.46 | 3.38 |
| LO | Lome | 6.13 | 1.22 |
| OG | Ogbomosho | 8.13 | 4.25 |
| PH | Port Harcourt | 4.82 | 7.03 |
| ZA | Zaria | 11.07 | 7.70 |



**Table 2.** Lognormal fit parameters for the fits shown in Fig. 8, using the equation $dN/dlnD_\mathrm{p} = \frac{N}{\sqrt{2\pi}ln(\sigma)} exp\left(-\frac{(ln(D_\mathrm{p})-ln(D_\mathrm{med}))^2}{2ln(\sigma)^2}\right)$.

|  | $N(\mathrm{cm}^{-3})$ | $\sigma$ | $D_\mathrm{med}$ (nm) |
|---|---|---|---|
| Offshore wide Aitken mode | 282 | 1.65 | 75.5 |
| Offshore accumulation mode | 570 | 1.50 | 199.1 |
| Local Aitken mode | 2829 | 1.66 | 56.5 |
| Local accumulation mode | 175 | 1.42 | 242.5 |



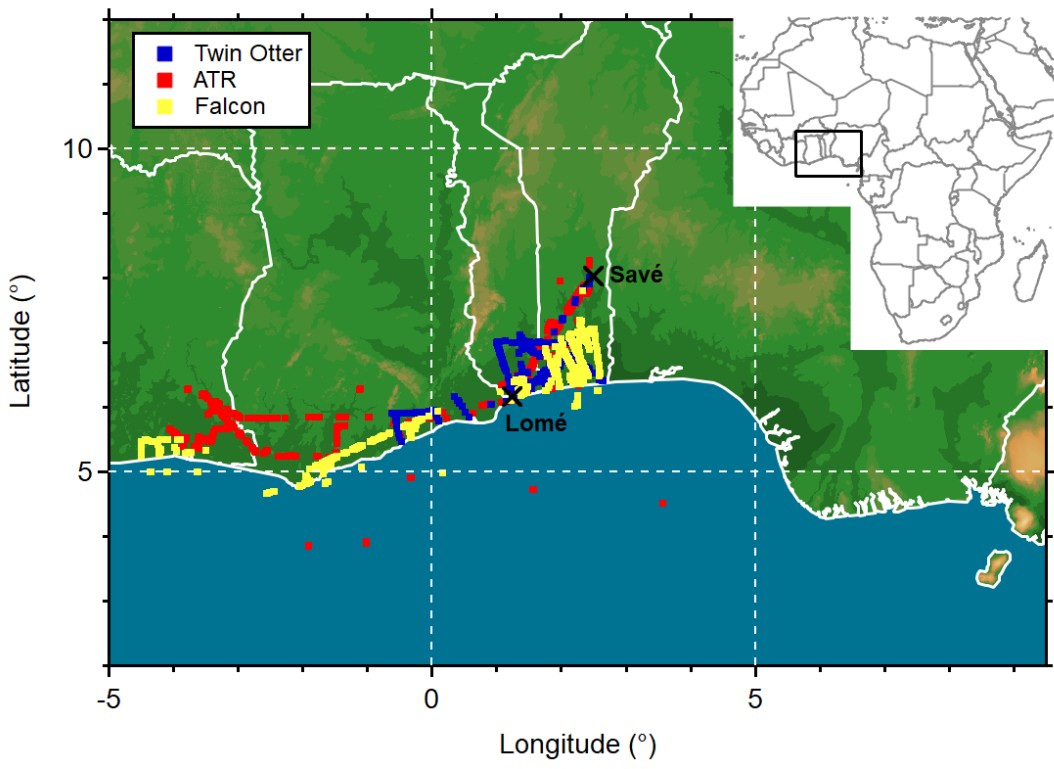

**Figure 1.** Map showing the location of all cloud measurements made below 1 km above mean sea level. The locations of Lomé and Savé are marked as many research flights took place between these two locations.





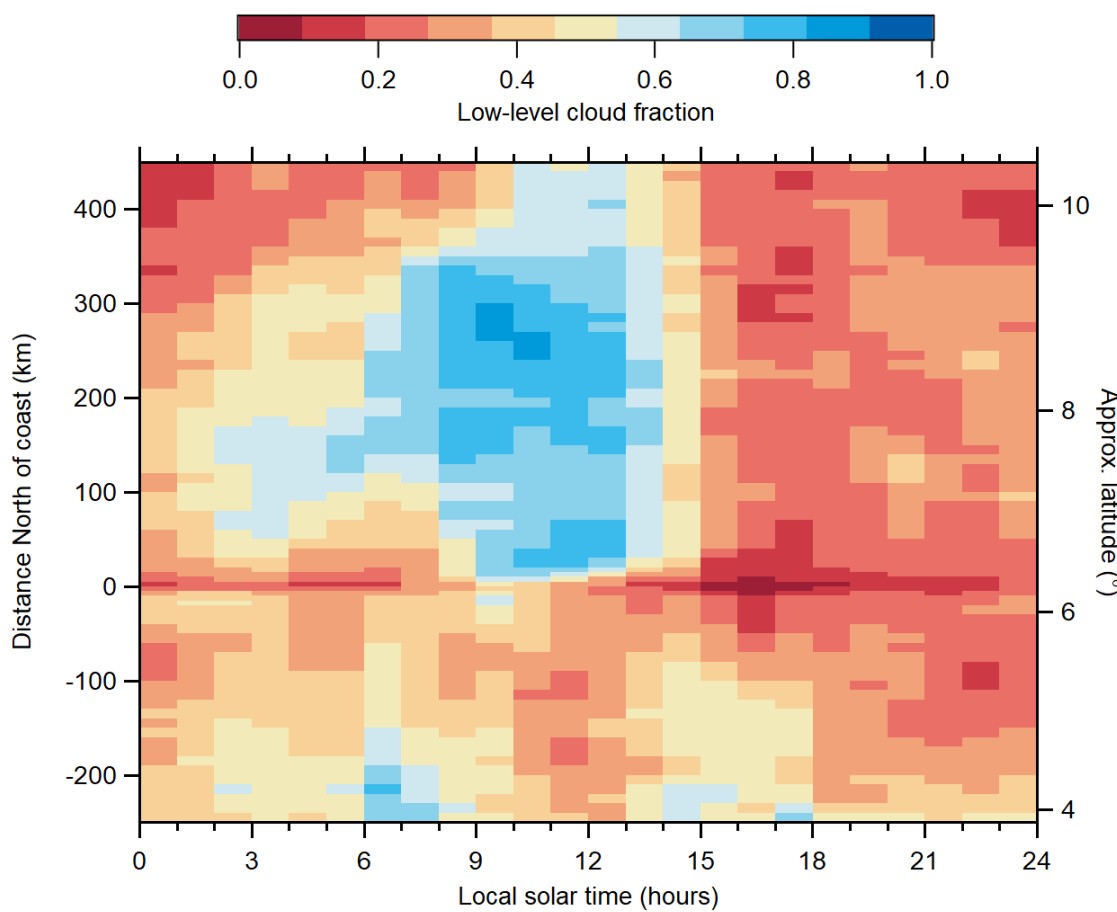

**Figure 2.** The diurnal cycle of mean low-level cloud fraction at different distances inland or offshore over Togo & Benin (4 – 11°N, 0.6 – 2.75°E), taken from the LLC flag derived from SEVIRI cloud data. The data are shown from the time period coinciding with the DACCIWA aircraft campaign: 29th June – 16th July 2016. The latitude scale on the right axis is approximate, as the latitude of the coastline varies by 0.6° over Togo & Benin.



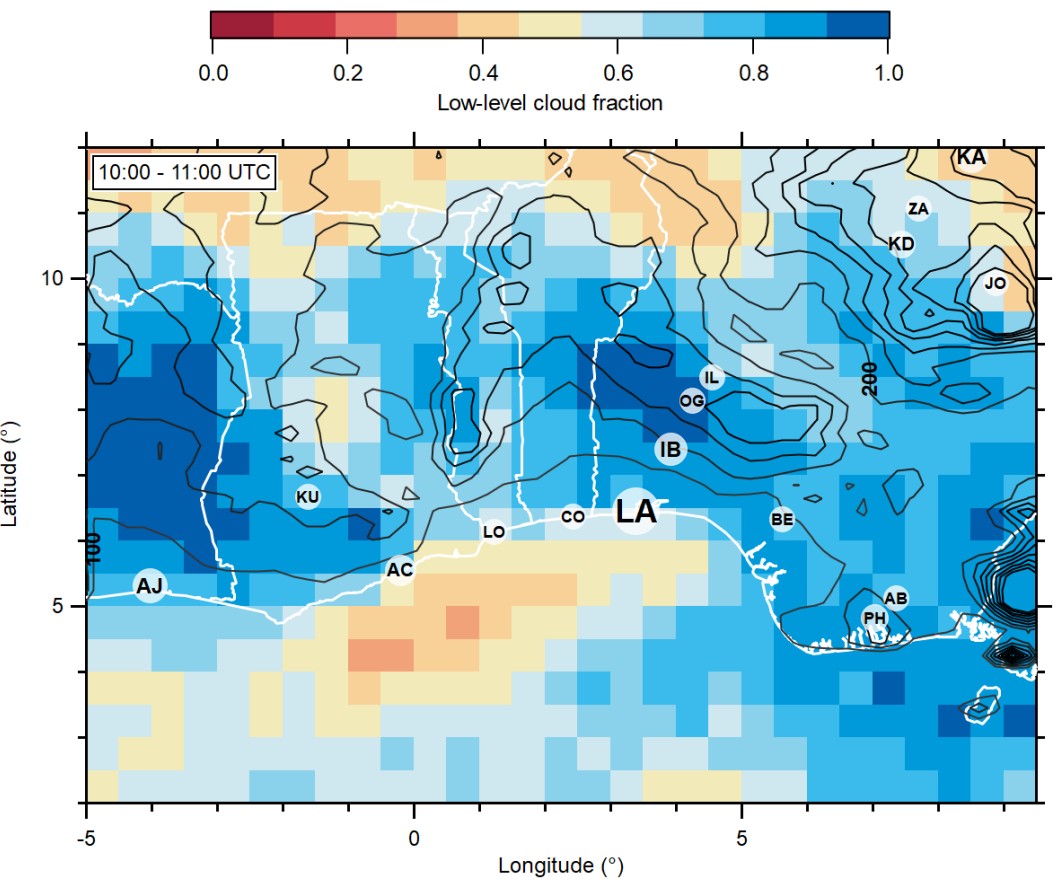

**Figure 3.** Mean low-level cloud fraction between 10:00 – 11:00 UTC during the DACCIWA aircraft campaign, 29th
June – 16th July 2016, taken from the LLC flag derived from SEVIRI cloud data. The contours show the elevation of the
land surface, and are labelled in metres above mean sea level. Text markers show the locations and abbreviated names of
large cities in the region, which are listed in Table 1. The text size gives a qualitative indication of the relative population
of each city.





**Figure 4.** Measured CDNC for all data below 1 km, stratified by distance inland and pollution conditions. Panel (a) shows all data as a function of distance from the coast. The markers and error bars show the median, 25th, and 75th percentiles. The numbers at the top show the number of different days' worth of data in each bin, as well as the number of individual data points. Panel (b) shows normalised histograms of CDNC at different distances from the coast. Panel (c) shows a histogram of in-cloud pollution, and panel (d) shows normalised histograms of inland CDNC stratified by CO concentrations.





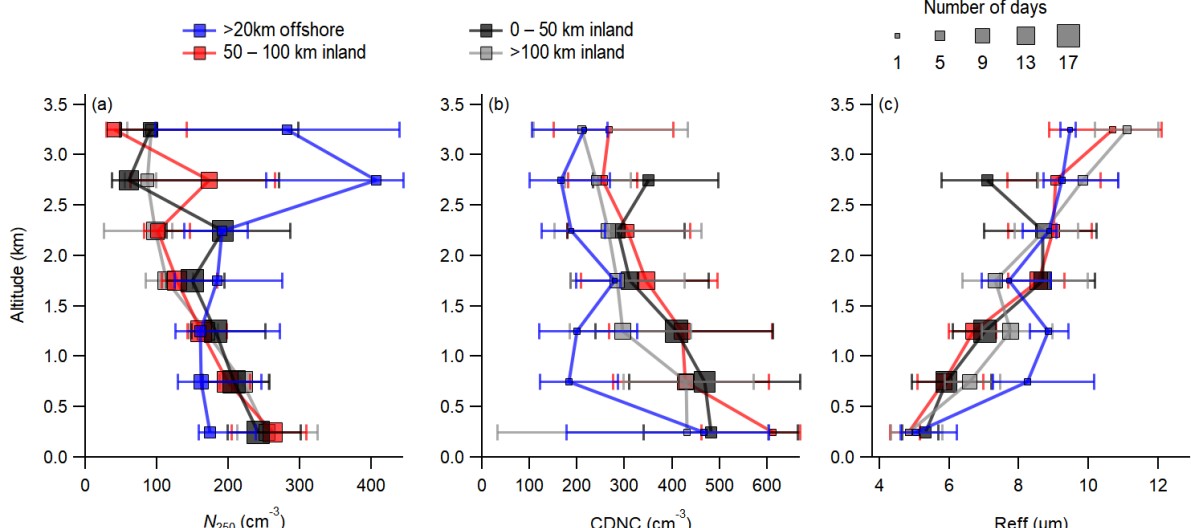

**Figure 5.** Average vertical profiles of aerosols larger than 250 nm (a), cloud drop number concentration (b), and effective radius. The markers are the medians, and are different sizes depending on the number of individual days the data are taken from. The error bars show the 25th and 75th percentiles, and the solid lines connect the medians.



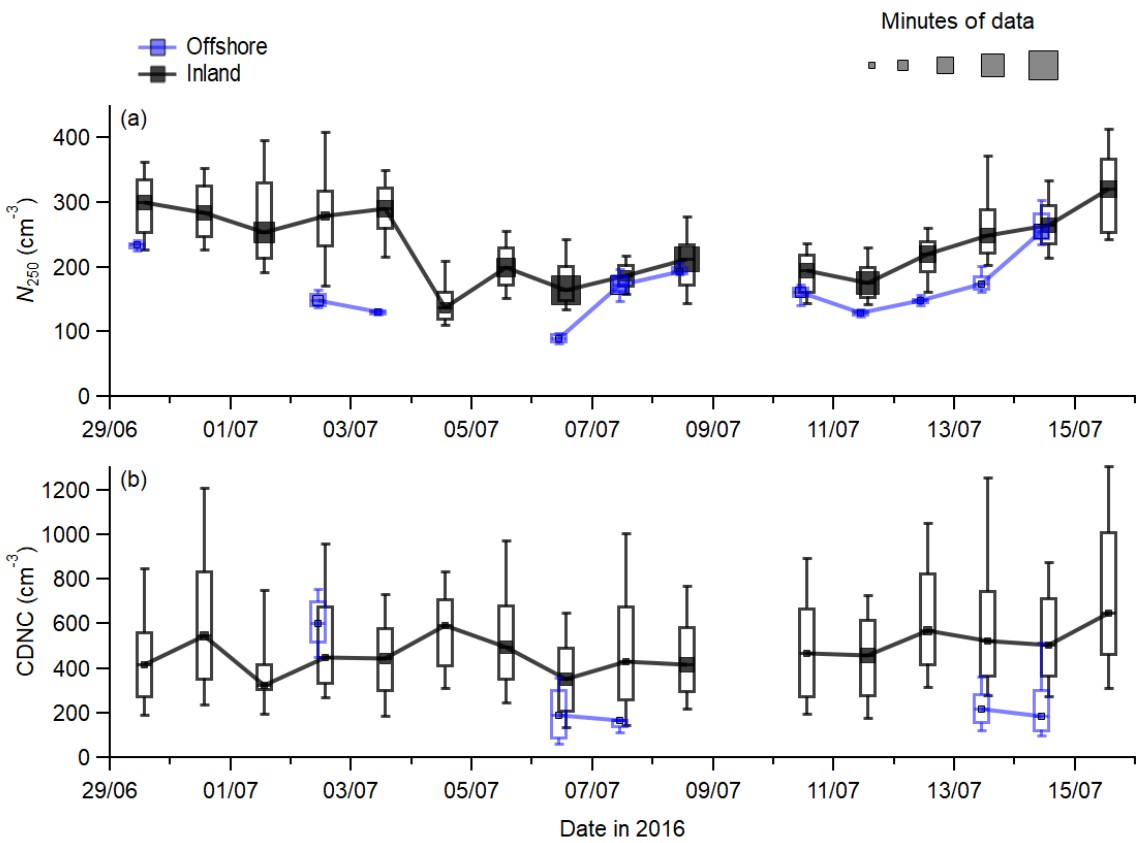

**Figure 6.** Day-to-day variation in $N_{250}$ (a) and CDNC (b). The data shown are from all data measured below 1km. The markers, boxes, and bars show the medians, 25th and 75th percentiles, and the 10th and 90th percentiles respectively. Offshore data were collected over 20 km south of the coast.



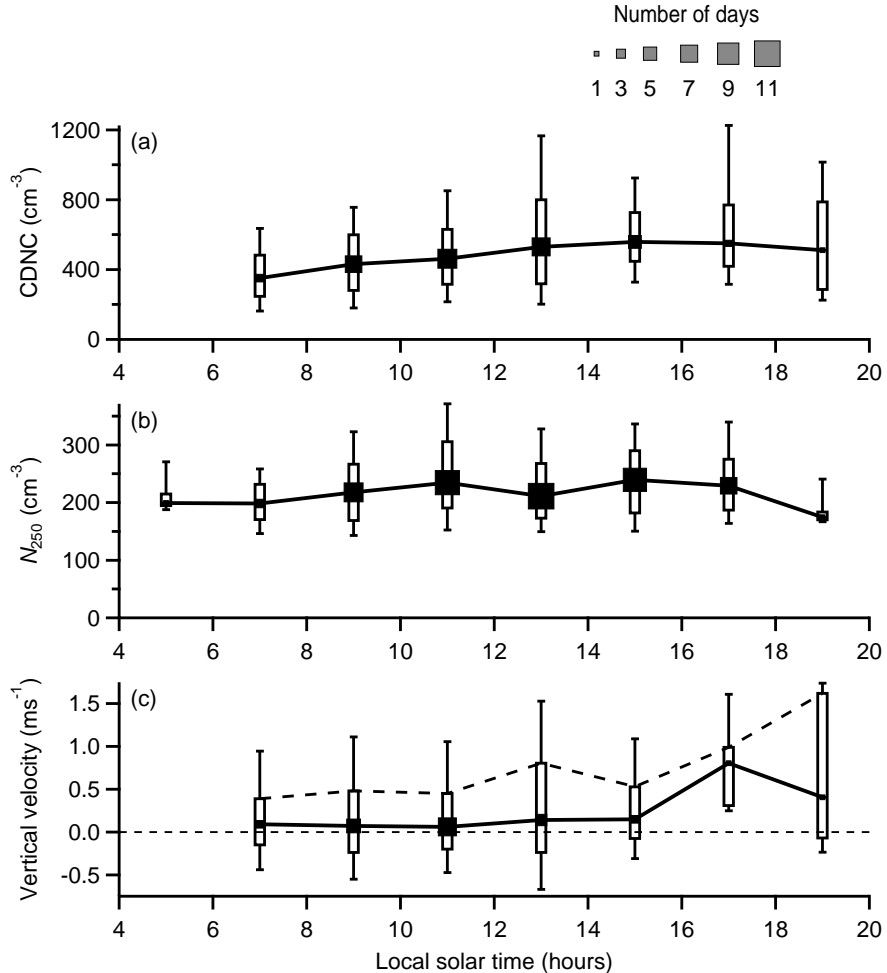

**Figure 7.** Inland diurnal variation in CDNC (a), $N_{250}$ (b), and in-cloud vertical velocity (c). The data shown are from all data measured below 1km. The markers are the medians, and are different sizes depending on how many individual days the data are taken from. The boxes are the 25th and 75th percentiles, and the bars are the 10th and 90th percentiles. The solid lines connect the medians, and the dashed line in part (c) connects the 75th percentiles.



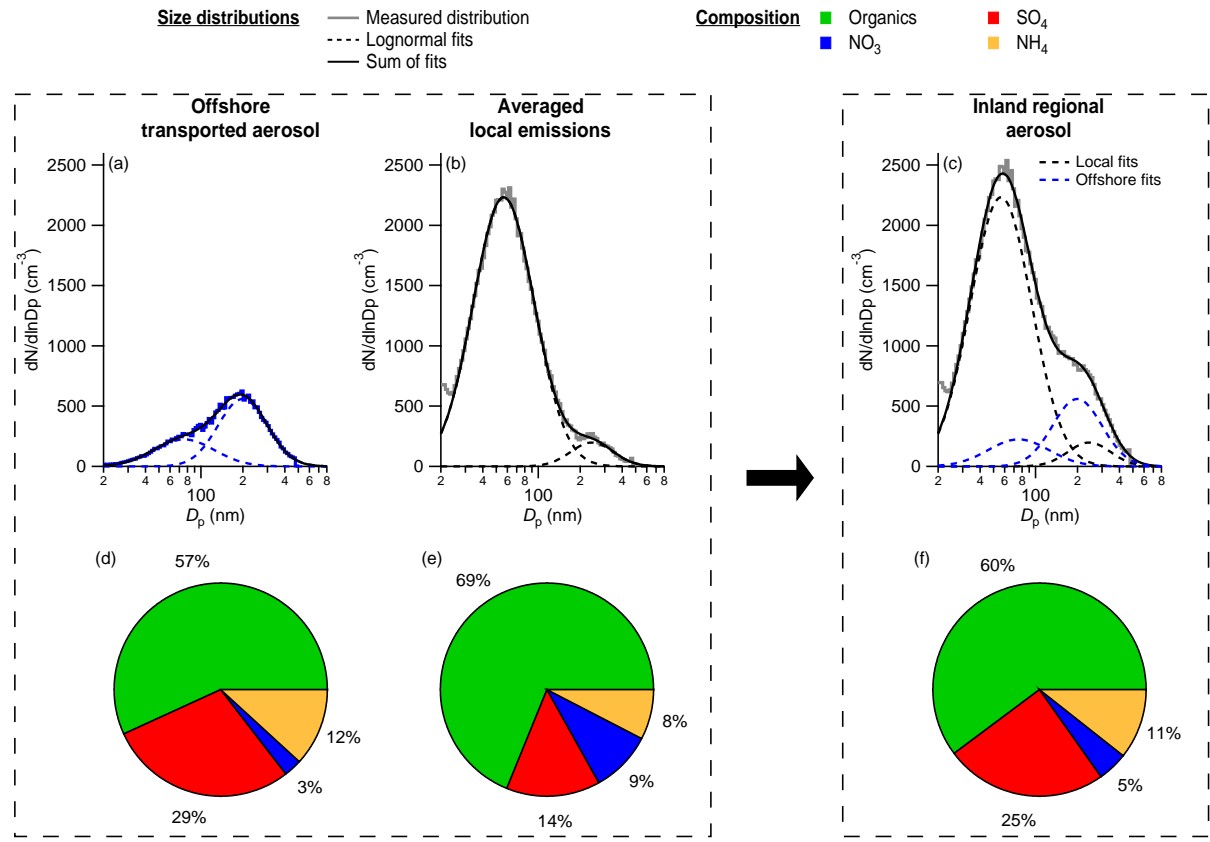

**Figure 8.** Aerosol size distributions (a – c) and accumulation mode composition measurements (d – f) made over SWA, that were used as input into ACPIM. The averaged local emissions were calculated by subtracting the offshore transported aerosol from the inland regional aerosol.





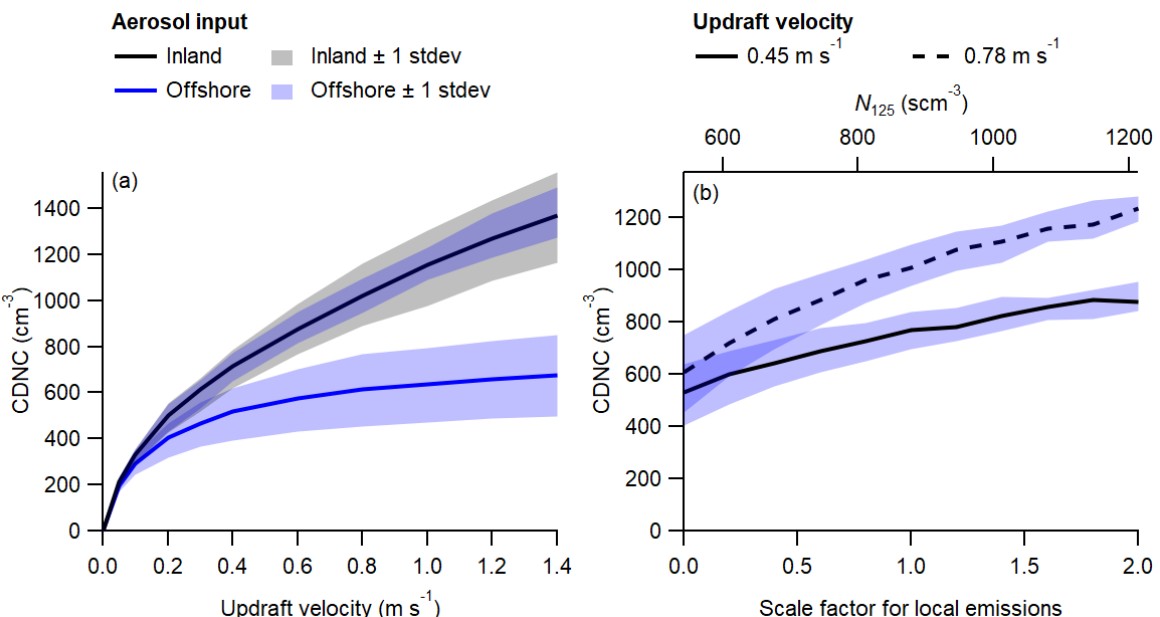

**Figure 9.** CDNC calculated during ACPIM parcel model runs initialised using measured aerosol size distributions and compositions shown in Fig. 8 and Table 2. The shaded regions show the expected day-to-day variability in CDNC as a result of variation in the aerosol concentrations, based on the standard deviation of measurements shown in Fig. 6a. Panel (a) shows the sensitivity of CDNC to updraft velocities, and panel (b) the sensitivity to different scale factors for local emissions in the aerosol input.