# Peer review of "Aerosol influences on low-level clouds in the West African monsoon"

_Atmospheric Chemistry and Physics, 2019_

## Referee Comment (RC1) · Anonymous Referee #1 · 11 Feb 2019

The authors use measurements from multiple aircraft to construct a large dataset of cloud properties. The authors show differences in these cloud properties based on location, and provide explanations for their differences based on pollution tracers and the understanding of the local dynamics and meteorological phenomenon. The authors conclude that significant increases in population in south west Africa and resulting enhanced pollution will not greatly affect cloud properties in the region, because a considerable fraction of aerosol particles are high concentrations of biomass burning smoke transported from the southern hemisphere. I have some concerns about the evidence being used to make this claim. I think the use of SMPS measurements would make a stronger case than OPC measurements alone. Furthermore, it is understandable that mixing state can be difficult to determine, but the expected hygroscopicities of

different particles are very different and simply assuming an internal mixture weakens the argument (aged biomass, fresh emissions, sea salt).

Major concerns:

A considerable amount of the introduction is attributed to explaining the effect of the low level jet on the inland clouds. While it is necessary for the reader to understand the role of the LLJ, I suggest limiting the explanation some since it is not the focus of the results. I would suggest performing a few parcel model simulations to identify how sensitive the simulation results are to mixing state, even if they are speculative. Proving to the reader that this (arguably bad) assumption cannot account for major changes in CDNC would be helpful to your results. Section 3.4 analysis with CDNC and Reff seems a little pointless, and possibly misleading. It's nice and simple to group up all your data, show the statistics and interpret them, but you're mixing together very different cases. You pointed out the fact that your profiles often contain multiple layers of clouds. And you pointed out that offshore measurements of CDNC are only for a day or so, but you still make plenty of comparisons despite this and I don't see the point. These offshore cloud measurements are certainly not enough to suggest they are representative of the regions offshore clouds and they are distracting from your real results/objectives.

Did you have a CPC? Figure 6 would benefit from such measurements. They include particles that are too small to be CCN, but N250 certainly misses a lot of CCN. CPC measurements would also provide some insight on the pollution levels.

Beginning of Page 14: Often the majority of the accumulation mode particles (and CCN) are smaller than 250 nm so based on the OPC data alone it's not entirely convincing that the offshore particles account for the majority of the accumulation mode. I understand there are limitations in the SMPS data, but for an analysis where you are averaging all concentration measurements over the land, it should not be a problem. Furthermore, I believe the next paragraph, where you compare CDNC between offshore and inland clouds to counter this conclusion with the inland CDNC often being more than double the offshore CDNC. Though again, there aren't quite enough off-shore clouds measured, but stating that inland clouds have 85-175% more droplets makes your argument less convincing. It's likely the different dynamics on and offshore may play a role here, so comparing the concentration of particles below cloud (from a CPC and SMPS) would be more useful.

Generally aerosol size distributions should not be averaged together unless they look similar (similar peak locations). Can you add all the distributions averaged in Figure 8 with light lines? It would also give the reader a good picture of the distribution variability.

Page 18 first paragraph: The argument on entrainment is a bit week. Sure you had a lot of layered clouds, but clouds with greater than 1 g mˆ-3 of LWC were often likely convective, and therefore, higher updraft velocities. At the very least I would include standard deviations for these CDNCs and the number of cases.

Minor points:

Page 8 line 15 change 'an' to 'and'.

Page 13 "This is"

Page 15 line 19. Do you mean scanning time?

---

## Referee Comment (RC2) · Anonymous Referee #2 · 22 Feb 2019

The presents and summarises cloud and aerosol observations in low-level clouds over the coast of southern West Africa made during the DACCIWA campaign. Further, an analysis of cloud cover in the wider area is presented based on satellite data and the impact of local aerosol emssions on cloud properties is investigated using a parcel model. The authors draw the conclusions that there is discernible impact of local emissions on the cloud properties and that changes in local emission strength would only have a small impact on CDNC and cloud properties.

In general, the paper is well written, the methods are mostly suitable, and the results are for the most part presented well and discussed logically. My main concerns regarding the main conclusions is a lacking discussion of whether the area porportion affected by local aeorosol emissions is reflected roughly correctly in the aircraft data.
Also in several places, it is not clear how the authors arrive at certain statements (s. comments below). If the necessary clarifications are included, I recommend to publish the paper.

**1  Specific comments**

1. p. 2, l. 4 ff: I believe the authors summarise the typical diurnal cycle during the campaign. This should be stated somewhere.

2. p.5, l. 18 & Fig. 1: The text states Fig. 1 is showing the flight tracks of the aircraft, while the caption says it shows the locations of cloud observations. Please clarify this inconsistency.

3. p. 5, l. 26: It would be helpful to state the actual number of days, for which observations over the sea are available here. Both for aerosol observations and cloud observations. Also can you clarify, how and why you arrive at the conclusion that the number of aerosol observations is sufficient to estimate offshore aerosol variability?

4. p. 6, l. 14 ff: For a better understanding of the relevance / meaning of the statistical comparisons, it would be good to include at least a short statement on the sampling times (hours of day), locations and strategies of the different aircraft.

5. p. 6, l. 21 ff: Can you also provide a comparison of the stastics of $R_{eff}$ other than the mean?

6. p. 7, l. 25: Is this the percentage difference in cloud fraction or the absolute difference in cloud fraction?

7. p. 8, l. 17 ff: The limitations of the modelling scheme are described. However, a discussion on how these assumption could impact the results is not included.

Could you at least speculate on the impact of the limitation on the results (maybe in the discussion section)?

8. p. 9, l. 15: It maybe would be interesting to have a figure similar to Fig. 2 with the standard deviation (or variance) in cloud fraction. This would allow a better judgement on, how robust the analysis of the mean field is.

9. p. 10, l. 5ff: Can you elaborate on how you determine whether there is or is not a clear effect on the cloud cover?

10. p. 11, l. 13: Is there some means of quantifying this fraction? Is it over- or underrepresented in the observational data? Do analysis of cloud microphysical profiles in the polluted region only?

11. p. 12, l. 27ff: I am a bit lost with the argumentation here. You say CDNC differences are "larger than differences in aerosol", "correlate with differences in the aerosol vertical profile" and are not "directly related to differences in accumulation mode aerosol"?

12. p. 13, l. 8ff: Are clouds over the ocean and the land multi-layered. If so the arguments regarding the connection between CDNC and $R_{eff}$ need to be considered more carefully, as with observations at the same altitude and varying cloud base altitudes this connection is not robust.

13. p. 13, l. 16 / 23: What is the basis for the claim of "little systematic spatial variability" and the statement of "fairly homogeneous" clouds inland?

14. p. 14, l. 5: What do you mean with a "degree of correlation"?

15. p. 14, l. 13: If this is solely based on Fig. 6b, I do not find the claim of an impact of the aerosol concentration on the dailz CDNC very convincing. Just from looking at the plot, I do not think there is any correlation in this data.

16. p. 15, l. 4: Can you provide any information regarding the position of the flight data relative to cloud base?

17. p. 18, l. 7: What part of the observed vertical velocity distribution are you using for this comparison? How does the observed vertical velocity relate to the cloud base vertical velocity?

18. section 3.1 & 3.2: It should be mentioned somewhere explicitly that the area anlaysed here is significantly larger than that covered by the aircraft observations. Orographic effects etc. discussed here have only a limited impact on the detailed cloud data.

19. Fig. 9: What does the blue shaded region around the inland line show?

**2   Technical corrections**

- p. 5, l. 23: " ... included **measuring** emissions ..." (?)
- p. 8, l. 15: "... updraft velocities **and** aerosol number ... "
- p. 12, l. 14: What is "it" refering to?
- p. 13, l. 5: I belief there is something missing from the sentence starting with "In the free troposphere ..."
- p. 13, l. 6: "This **is** due ..."
- p. 13, l. 8: Do you mean cloud observations at higher altitude were not necessarily higher above the cloud base than observations at lower altitude?
- p. 13, l. 9: "... which is **consistent** with ..."
- p. 14, l. 24: " ... indicating **a** greater cloud ..."

- p. 16, l. 28: "... updraft **than** in aerosol ..."
- p. 18, l. 5: "... measurements **of** aerosol composition ..."

---

## Referee Comment (RC3) · Anonymous Referee #3 · 27 Feb 2019

Review of "Aerosol influences on low-level clouds in theWest African monsoon" by Taylor et al., submitted to ACPD

The work describes insights on the low level cloud deck which forms during the Monsoon season over South West Africa. These were derived from aircraft measurements of number concentrations of both cloud droplets and aerosol particles in combination with satellite data and some modeling efforts.

It is generally well written, but it seems that data shown in figures does not always support the conclusions drawn. My major concern is about the conclusion that changes in aerosol particle number concentrations (APNC) do not effect cloud droplet number concentrations (CDNC). And I will not argue that there is a large effect, but the presented data also does not support that there is no effect.

[Figure]

I should say upfront that I started reading this paper not having any agenda of my own on this topic. While writing this report, time and again I saw hints in the manuscript which support an influence of aerosol particle number concentrations (APNC) on cloud droplet number concentrations (CDNC). But a contrasting interpretation was given, arguing that the amount of data for the "low concentrations cases" (i.e., offshore) is too small for interpretations. In the following, I will comment on a number of related text passages and on data shown in your figures which seem to indicate a different result. Your work needs to account for that.

Nevertheless, the topic is timely and a revised version may merit publication in ACP.

Major concerns:

As said above, this major concern deals with a possible effect of APNC on CDNC. I understand that you try not to over-interpret the (comparably small number of) data you got offshore. But still, I see the same there, over and over again: Cloud droplet number concentrations (CDNC) are lower offshore than inland, and so are aerosol particle number concentrations (APNC, in your case particles larger than 250nm (N_250). And also the aerosol particle size distribution (APSD) you derived for the modeling is clearly lower in number and different in shape offshore, compared to inland.

If I understood it correctly, you argue that even offshore the air masses are always polluted, originating from biomass burning aerosol that prevails in this region during this time of the year (e.g., page 21, line 24-25). This is a bit confusing, as you also describe that you went > 20km offshore "to unambiguously remove any possibility of local terrestrial emissions affecting" (page 10, line 25-26). So this would mean that the pollution from biomass burning aerosol should be ubiquitous in the whole region, also > 20km offshore. But the size distribution you show as representative for > 20km offshore (Fig. 8) does not indicate that biomass burning pollution is prominently present (for that a pronounced mode of particles < 100nm would be expected). So I somewhat wonder how "polluted" air masses in the offshore region really are.

[Figure]

Generally, you omit the lower concentrations offshore and if you refer to them at all, you say that these data are not to be trusted, due to their scarcity. It is true that they are scarce, but if you really do not trust them, you should not use them in the first place. This is NOT what I would recommend. But I would still recommend that you spend some thoughts on the idea that there might be a real increase in APNC as the air masses go over land, and related to that an increase in CDCN. And I also recommend that you adjust your interpretations in the text.

Below I collected the passages in your manuscript I stumbled upon, which all more or less deal with this issue:

On page 10, line 4-5, referring to Fig. 3, you say "Despite the presence of urban emissions, anthropogenic aerosols had no clear effect on cloud cover downwind of the major cities, . . .". This seems vague - by simply looking at Fig. 3, one region with the highest LLC fraction (in the middle of the figure, longitude 3-4°, latitude 8-9°) is north of the biggest city you indicated (LA) and also close to the highest "density" of cities on the map (IB, OG, IL). And then there is another high LLC fraction north of AJ, another larger city. - At the same time, there are mountain slopes, so you might be correct, but judging from the figure you cannot say that anthropogenic aerosol did not have a clear effect. (I wrote this remark given above before reading on, and later found more on this:) You do show later on that there is an effect (Fig. 4d) on CDNC (which is, on average, increased to $\sim$ 1.5 times higher values, page 12, line 2), and you again show this in your modelling – so there might not be a large effect, but still, there is one, so your remark on page 10, line 4-5 should be tuned down, maybe already pointing towards these later results, or deleted at all. And I think it is all the more interesting that you see this increase in the CDND in Fig. 4d ALTHOUGH "However, these highly polluted plumes represent less than 10% of the inland data measured over the region." (as you say on page 12, lin 2-3).

page 11, line 10: Already upon my first reading, I made the following note here, and this got only more relevant: The most striking difference between offshore and inland

data, as I see it, is the larger average CDNC for the inland data. You do not discuss this at all. Would there be generally larger aerosol concentrations over land (also from natural sources)? Or any other idea where that could come from?

page 12, line 14-15: Here you clearly describe that the ground acts as an aerosol source: "Inland, accumulation mode aerosol concentrations were highest closer to the ground, and decreased with altitude up to around $2 – 2.5$ km." There is a factor of roughly 2.5 between values of $N\_250$ on ground and above 2km.

page 13, line 1-2: When looking at Fig. 5, it struck me that up to 2.5km mostly $N\_250$ and CDNC are roughly similar for the offshore cases, while for the inland cases CDNC are roughly double the amount of $N\_250$. That would imply that there must be more particles < 250nm in the air inland (compared to offshore) which can be activated to cloud droplets, otherwise the CDNC are difficult to explain.

page 13, line 14ff: You say "There are differences in aerosol and CDNC between inland and offshore clouds, but not in $R\_Eff$, . . . " – below 1.5km, the "bump" that shows up in CDNC for the offshore case (towards lower values at 0.75 km and 1.25 km) is reflected in larger $R\_Eff$ (comparing $\sim$ 8-9 micrometer offshore to $\sim$ 6-7 micrometer inland). So this sentence needs a reformulation. Already Fig. 4b showed a totally different cloud droplet size distribution for offshore and inland conditions.

page 14, line 2-3: There is a factor of $\sim$ 1.5 between $N\_250$ that you give for offshore and inland, again with concentrations inland being larger. This, together with my comment concerning page 13, line 1-2, I am not sure if it can really be said that "aerosol imported from offshore comprised the majority of accumulation mode aerosol inland".

page 15, line 8-10: Increase in mean updraft and in CDNC do not really mirror each other. While CDNC increase from 7 to 13 LST and then stay quite stable, the vertical velocity, implying a possibly stronger role of dynamics, is quite stable and has a distinct peak at 17 LST (followed by data ta 15 and 19 LST). So it is not so clear to me that dynamics explain the observations.

[Figure]

page 15, line 23-25: It would be good to know how the inland size distribution exactly was derived (measured when and where)? But in any case, here it can clearly be seen that inland and offshore particle number size distributions differ massively, with much higher concentrations for particle sizes $\sim$ < 200 nm for the inland aerosol.

page 19, line 21-25: As said above, looking at your data gave me an impression that is deviating from your interpretation. Instead, and increase in particle number concentrations from offshore to inland conditions might indeed play a role.

page 19, line 21-23: "It may be that our poor statistics of offshore data present a misleading distribution in Fig. 4b, or that there was a systematic and ubiquitous increase in CDNC due to some other factor such as stronger updrafts inland." Again, the "poor statistics" offshore is certainly better constrained than stronger updrafts inland. Admittedly, an increase by a factor of 1.5 for CDNC might not be huge, but at least there seems to be some effect, there.

page 20, line 23-24: In light of all the above, this sentence should be revised.

page 21, line 1-3: And again, there is evidence in your data that APNC increased as soon as air masses were transported over land, although they might have been already "mildly polluted". (The particle number size distributions offshore and inland are VERY different!) And it is true that polluted air masses are less susceptible to a further increase in APNC, but you do see an increase, nevertheless. So again, this sentence should be revised.

page 21, line 24-25: Again, the "high aerosol background from transported smoke" was somewhat lower than what that sentence suggests.

Minor and technical comments:

page 9, line 11-13: You write "A further region of higher cloud cover was seen just inland . . . ". If I get the color scale right, then closer to the coast the coverage is 70%, drops to 60% and then again increases to 70%. The description here sounds a bit

different. To me this area does not necessarily look like two separate regions, and the clouds closer to the sea do not really dissipate as they move further inland. A reformulation is needed.

page 13, line 6: "is" is missing between "This due".

page 13, line 12: The sentence is more general than what you report. Following "This implies that offshore clouds" insert "at altitudes above 1.5 km" and in the following line change "that clouds offshore" to "that these offshore clouds".

page 14, line 18-19: The CDNC does not really "peak around 15:00 LST". There is a clear increase from before to after local noon, but after that, looking at all mean and percentiles, it seems to rather reach a plateau, until after 18:00.

page 17, line 1-2: You say ". . . for the average updrafts the main sensitivity of CDNC is not so clear-cut." What do you mean by "is not so clear-cut"? For the average updrafts, there is a difference, as you state in the text above, so I am confused by this statement here.

page 17, line 3-4: You say that "A large fraction of the day-to-day variability . . . derives from the day-to-day variability in the offshore aerosol brought inland." But isn't this indicated by the bluish color band? The variability of the inland aerosol (greyish color band) is much larger. This confused me a little (I thought you might have mixed up the colors, but maybe you didn't?), and it would be fair to at least mention the fact that the inland aerosols' variability is even larger.

page 17, line 4: Following the above mentioned sentence, you mention "these average updrafts", and maybe I was still confused from that "color issue" above, but I first didn't get which updrafts you mean. Maybe it would be better to relate to them as "the average morning and afternoon updrafts as defined above", or something like this.

page 17, line 18-19: The change in CDNC you report for omitting local emissions in your model also nicely reflects the change in CDNC between offshore and inland that

you showed above. You should mention this.

page 17, line 20ff: "The effect of varying local aerosol emissions does not produce a linear relationship between CDNC and total aerosol concentrations." One additional reason for this, which you do not mention explicitly, yet, is that there is this rivalling effect where the maximum super-saturation at a given updraft is lower when there are higher concentrations of CCN, due to the water vapor being consumed faster, hence the critical size above which particles activate to droplets becomes larger.

page 18, line 21: Is "flat" a good way to describe an ocean surface? Maybe it is (I'm not a native speaker), but I would think there might be a better word, like "calm" or something else?

page 22, line 22: There are too many quotation marks around "borderline".

---

## Author Comment (AC1) · 30 Apr 2019

**Reviewer #1**

**The authors use measurements from multiple aircraft to construct a large dataset of cloud properties. The authors show differences in these cloud properties based on location, and provide explanations for their differences based on pollution tracers and the understanding of the local dynamics and meteorological phenomenon. The authors conclude that significant increases in population in south west Africa and resulting enhanced pollution will not greatly affect cloud properties in the region, because a considerable fraction of aerosol particles are high concentrations of biomass burning smoke transported from the southern hemisphere. I have some concerns about the evidence being used to make this claim. I think the use of SMPS measurements would make a stronger case than OPC measurements alone. Furthermore, it is understandable that mixing state can be difficult to determine, but the expected hygroscopicities of different particles are very different and simply assuming an internal mixture weakens the argument (aged biomass, fresh emissions, sea salt).**

We thank the reviewer for their useful comments and will address them individually below.

**Major concerns:**

**A considerable amount of the introduction is attributed to explaining the effect of the low level jet on the inland clouds. While it is necessary for the reader to understand the role of the LLJ, I suggest limiting the explanation some since it is not the focus of the results.**

We thank the reviewer for the suggestion. We have shortened the paragraph about the role of the LLJ.

**I would suggest performing a few parcel model simulations to identify how sensitive the simulation results are to mixing state, even if they are speculative. Proving to the reader that this (arguably bad) assumption cannot account for major changes in CDNC would be helpful to your results.**

We have added to section 4.2

*"We performed some additional simulations to explore some possible sensitivities in our assumptions. Firstly we considered the presence of a coarse sea salt aerosol mode, which we isolated from the GRIMM size distributions. The size distribution of this mode is listed in Table 2, and the composition was assumed to be NaCl. As the concentration in the sea salt mode is so low compared to the accumulation mode, this addition of a coarse sea salt mode made less than a 1% difference to the derived CDNC. We also performed some illustrative runs on the inland aerosol- one possible scenario is that the local Aitken mode aerosol were from primary combustion. We assumed them to be 100% organic, and proportioned the composition of the local accumulation mode to match the total measured organic/inorganic ratio. Compared to the base case, this increased the inorganic*

*fraction in the local accumulation mode from 31% to 41%, and the resultant CDNCs showed around a 12% increase. These simulations show that there are some differences that are attributable to assumptions in aerosol composition and mixing state, but they are likely to be smaller than the differences we show related to the total amount of aerosol and updraft velocity."*

**Section 3.4 analysis with CDNC and Reff seems a little pointless, and possibly misleading. It's nice and simple to group up all your data, show the statistics and interpret them, but you're mixing together very different cases. You pointed out the fact that your profiles often contain multiple layers of clouds.**

We think it's worth providing these statistics to provide a basis of comparison for models. If a model is not producing average properties like these then it is not accurately reproducing the microphysical properties of the clouds. We have added the caveat to the caption to Figure 5

*"The cloud profiles show the average properties of the cloud field, which often contained multiple discreet layers."*

In the text relating to Figure 5c (the Reff profile) we already explicitly discuss the presence of multiple cloud layers.

**And you pointed out that offshore measurements of CDNC are only for a day or so, but you still make plenty of comparisons despite this and I don't see the point. These offshore cloud measurements are certainly not enough to suggest they are representative of the regions offshore clouds and they are distracting from your real results/objectives.**

For the offshore cloud measurements it seems strange to omit them entirely, given we rely on the offshore aerosol measurements for the basis of our analysis. Reviewer #3 has also explicitly said they think we should not remove them. In every point in the results where they are discussed, we have already noted that they are limited in scope. They do also support the big picture argument we are trying to make, that the polluted background aerosol is having an impact on the clouds. We have made significant changes to the discussion in response to this and Reviewer#3's significant comments on this issue

**Did you have a CPC? Figure 6 would benefit from such measurements. They include particles that are too small to be CCN, but N250 certainly misses a lot of CCN. CPC measurements would also provide some insight on the pollution levels.**

Haslett et al (2019) provide CPC measurements for the offshore/inland data. In the urban outflow regions, large numbers of Aitken mode particles are present, that are too small to act as CCN. We thought would confuse the issue given we are interested in CCN, and the accumulation mode aerosols show a very different trend. Also see below regarding the comparison to the SMPS.

**Beginning of Page 14: Often the majority of the accumulation mode particles (and CCN) are smaller than 250 nm so based on the OPC data alone it's not entirely convincing that the offshore particles account for the majority of the accumulation mode. I understand there are limitations in the SMPS data, but for an analysis where you are averaging all concentration measurements over the land, it should not be a problem. Furthermore, I believe the next paragraph, where you compare CDNC between off-shore and inland clouds to counter this conclusion with the inland CDNC often being more than double the offshore CDNC. Though again, there aren't quite enough off-shore clouds measured, but stating that inland clouds have 85-175% more droplets makes your argument less convincing. It's likely the different dynamics on and offshore may play a role here, so comparing the concentration of particles below cloud (from a CPC and SMPS) would be more useful.**

Regarding SMPS data, we have added this to section 3.5

*"The average inland enhancement is in excellent agreement with Haslett et al. (2019), who also found a 40% enhancement in accumulation mode aerosol in continental background areas compared to upwind marine, using SMPS measurements averaged over the whole campaign. The day-to-day averages of the accumulation mode aerosol number concentration from the SMPS showed good correlation with the GRIMM, with R 2 = 0.7, but data were only available for 9 days inland and 2 days offshore. The GRIMM data therefore show a similar trend as the SMPS, but with greater coverage."*

We have also added to the to the discussion

*"Our measurements showed an average ~40% increase in regional accumulation mode aerosol inland compared to offshore. The accumulation mode is the most important for CCN, as particles smaller than this are too small to activate. Haslett et al. (2019) also found a 40% enhancement in the inland accumulation mode aerosol concentrations compared to offshore using SMPS data, which is in excellent agreement with our GRIMM measurements of $N_{250}$ . In localised city plumes, highly concentrated emissions had a noticeable increase in CDNC, but on a regional scale the majority of accumulation mode aerosol were due to long-range transport."*

**Generally aerosol size distributions should not be averaged together unless they look similar (similar peak locations). Can you add all the distributions averaged in Figure 8with light lines? It would also give the reader a good picture of the distribution variability.**

This is a good suggestion to show the variability in the size distributions, and this plot can be found in Haslett et al (2019). We have added a clarification to section 4.1

*"While the Aitken mode was variable inland due to highly localised anthropogenic emissions, the accumulation mode showed minimal variability as the majority of particles in this size range were due to long-range transport (Haslett et al., 2019).*

*This averaging approach is therefore reasonable to inform cloud activation modelling, as the accumulation mode is the source of CCN"*

**Page 18 first paragraph: The argument on entrainment is a bit week. Sure you had a lot of layered clouds, but clouds with greater than 1 g mˆ-3 of LWC were often likely convective, and therefore, higher updraft velocities. At the very least I would include standard deviations for these CDNCs and the number of cases.**

We have removed this part of the paragraph, which we agree was a bit speculative. We have replaced it with

*"The values of modelled CDNC are between the 75th and 90$^{th}$ percentiles of measured CDNC shown in Fig. 7a, meaning the difference between the average model and measured values is smaller than the variability in the measured data. ACPIM is an idealised simulation and we have used it here primarily to explore the sensitivity to local aerosol emissions, rather than arbitrarily try to tune factors such as aerosol mixing state, more realistic dynamics, or entrainment (of which we have limited or no measurements) to make the CDNC agree perfectly."*

**Minor points:**

**Page 8 line 15 change 'an' to 'and'.**

Done

**Page 13 "This is"**

Done

**Page 15 line 19. Do you mean scanning time?**

Done

**Reviewer #2**

**The presents and summarises cloud and aerosol observations in low-level clouds over the coast of southern West Africa made during the DACCIWA campaign. Further, an analysis of cloud cover in the wider area is presented based on satellite data and the impact of local aerosol emssions on cloud properties is investigated using a parcel model. The authors draw the conclusions that there is discernible impact of local emissions on the cloud properties and that changes in local emission strength would only have a small impact on CDNC and cloud properties. In general, the paper is well written, the methods are mostly suitable, and the results are for the most part presented well and discussed logically. My main concerns regarding the main conclusions is a lacking discussion of whether the area proportion affected by local aerosol emissions is reflected roughly correctly in the aircraft data.**

**Also in several places, it is not clear how the authors arrive at certain statements (s.comments below). If the necessary clarifications are included, I recommend to publish the paper.**

We thank the reviewer for their useful comments, which we will address individually below.

**1 Specific comments**

**1. p. 2, l. 4 ff: I believe the authors summarise the typical diurnal cycle during the campaign. This should be stated somewhere.**

The abstract first paragraph now reads

*"This novel dataset allows us to assess spatial and temporal , diurnal, and day-to-day variation in the properties of these clouds over the region."*

**2. p.5, l. 18 & Fig. 1: The text states Fig. 1 is showing the flight tracks of the aircraft, while the caption says it shows the locations of cloud observations. Please clarify this inconsistency.**

We've clarified this caption, it's the parts of the flight track where cloud measurements were made

**3. p. 5, l. 26: It would be helpful to state the actual number of days, for which observations over the sea are available here. Both for aerosol observations and cloud observations. Also can you clarify, how and why you arrive at the conclusion that the number of aerosol observations is sufficient to estimate offshore aerosol variability?**

The number of days is shown in figures 5 and 6- in Figure 6 it is easy to see. We've clarified in the text, changing "several days" to *"11 days"*. It seemed reasonable that this would give us some understanding of the variability throughout the campaign, as we have >50% coverage and it is fairly evenly spread, as shown in Figure 6.

**4. p. 6, l. 14 ff: For a better understanding of the relevance / meaning of the statistical comparisons, it would be good to include at least a short statement on the sampling times (hours of day), locations and strategies of the different aircraft.**

We have taken effort to minimise the influence of any differences in sampling locations/strategies by limiting this comparison to a small geographical area and altitude <1km. This is already stated in the text.

**5. p. 6, l. 21 ff: Can you also provide a comparison of the stastics of $R_{eff}$ other than the mean?**

This now says

*"For $R_{Eff}$ , average vertical profiles between 0 and 100 km inland showed the median values agreed within ~1 µm, which is within the uncertainties of the instruments, and the interquartile ranges were similar for each platform."*

**6. p. 7, l. 25: Is this the percentage difference in cloud fraction or the absolute difference in cloud fraction?**

We've added that it's absolute difference

**7. p. 8, l. 17 ff: The limitations of the modelling scheme are described. However, a discussion on how these assumption could impact the results is not included.**

**Could you at least speculate on the impact of the limitation on the results (maybe in the discussion section)?**

We have addressed this is our addition to section 4.2 with the extra model runs, in response to reviewer #1.

**8. p. 9, l. 15: It maybe would be interesting to have a figure similar to Fig. 2 with the standard deviation (or variance) in cloud fraction. This would allow a better judgement on, how robust the analysis of the mean field is.**

We have added this as a part (b) to figure 2, and added the following text to section 3.2

*"The areas of high mean LLC are the most robust features, with the lowest standard deviation, meaning patches of continuous cloud were present on almost all days during the campaign. Areas with lower average LLC fraction had higher standard deviation and were therefore more variable; the cloud cover in these regions varied day-to-day but was lower on average."*

**9. p. 10, l. 5ff: Can you elaborate on how you determine whether there is or is nota clear effect on the cloud cover?**

We have added to section 3.2

*"Pollution plumes are expected to extend for hundreds of kilometres downwind of the major cities (Deroubaix et al., 2019). Figure 3 does not show any features extending*

*downwind of the major cities that would represent a signature for anthropogenic influence on cloud cover on a regional scale"*

**10. p. 11, l. 13: Is there some means of quantifying this fraction? Is it over- or underrepresented in the observational data? Do analysis of cloud microphysical profiles in the polluted region only?**

We have added that the CO threshold of 160ppbv is the 93$^{rd}$ percentile from our measurements, and also added

*"Our measurements in this polluted tail were generally in and around the major cities, which were the one of the research themes of the DACCIWA campaign. On the regional scale these are likely to be overrepresented compared to the more sparsely populated regions such as Ghana and Côte d'Ivoire, but may be underrepresented compared to the more polluted regions of coastal Nigeria."*

**11. p. 12, l. 27ff: I am a bit lost with the argumentation here. You say CDNC differences are "larger than differences in aerosol", "correlate with differences in the aerosol vertical profile" and are not "directly related to differences in accumulation mode aerosol"?**

This was a typo- "did correlate with differences in aerosol" should have been "did not correlate with differences in aerosol". Essentially we see differences in CDNC but they are larger in magnitude than the differences in aerosol and don't correlate with them, so it's not obvious that the variation in CDNC is due to aerosol. This is addressed in the updated discussion.

**12. p. 13, l. 8ff: Are clouds over the ocean and the land multi-layered. If so the arguments regarding the connection between CDNC and R$_{eff}$ need to be considered more carefully, as with observations at the same altitude and varying cloud base altitudes this connection is not robust.**

Please see our response to Reviewer #1's comment starting "Section 3.4"

**13. p. 13, l. 16 / 23: What is the basis for the claim of "little systematic spatial variability" and the statement of "fairly homogeneous" clouds inland?**

If you look at Figure 5, the medians for both aerosol and cloud properties for most alititude bins look almost the same for all 3 inland traces. Systematic spatial variability would be something like decreasing CDNC further inland, which the data do not show. The reviewer's "fairly homogeneous" statement here is a bit unfair- the full quote is *"Although the properties of individual clouds were variable, the average inland clouds below 1 km were fairly homogeneous regardless of the distance inland"*, which is true based on Figure 5 and the above.

**14. p. 14, l. 5: What do you mean with a "degree of correlation"?**

We've added that there was R$^2$ = 0.72 between the medians for CDNC and N$_{250}$ for the time period in question. Admittedly it is difficult to see from eyeballing the plot as the CDNC was so variable locally, we have added

*"On a cloud-to-cloud basis this correlation was dwarfed by the inherent local variability in CDNC."*

**15. p. 14, l. 13: If this is solely based on Fig. 6b, I do not find the claim of an impact of the aerosol concentration on the dailz CDNC very convincing. Just from looking at the plot, I do not think there is any correlation in this data.**

See the previous comment

**16. p. 15, l. 4: Can you provide any information regarding the position of the flight data relative to cloud base?**

See the next comment

**17. p. 18, l. 7: What part of the observed vertical velocity distribution are you using for this comparison? How does the observed vertical velocity relate to the cloud base vertical velocity?**

For this and the previous comment, we have added in to section 3.6

*"Vertical velocity does not vary strongly with altitude in stratocumulus clouds (Wood, 2012), so although our measurements are not just from cloud base, they are still suitable to use in our simulations."*

We've also clarified in section 4.2 that we were referring to the simulations *"using the representative morning and afternoon updrafts"*

**18. section 3.1 & 3.2: It should be mentioned somewhere explicitly that the area anlaysed here is significantly larger than that covered by the aircraft observations. Orographic effects etc. discussed here have only a limited impact on the detailed cloud data.**

This is a good point to make- we have added near the start of section 3.3

*"The limited spatial area of the aircraft sampling means that regional scale orographic effects discussed in Sect. 3.2 are not likely to have a large effect on in situ cloud properties."*

**19. Fig. 9: What does the blue shaded region around the inland line show?**

That's the variability in CDNC that you get from varying the offshore aerosol modes by +- 1 stdev, as labelled in the figure key.

**2 Technical corrections**

**• p. 5, l. 23: " ... included measuring emissions ..." (?)**

Done

**• p. 8, l. 15: "... updraft velocities and aerosol number ... "**

Done

**• p. 12, l. 14: What is "it" refering to?**

Clarified in the paper as *"$N_{250}$ does not show…."*

**• p. 13, l. 5: I belief there is something missing from the sentence starting with "In the free troposphere ..."**

We've clarified it's $R_{Eff}$ we're talking about

**• p. 13, l. 6: "This is due ..."**

Done

**• p. 13, l. 8: Do you mean cloud observations at higher altitude were not necessarily higher above the cloud base than observations at lower altitude?**

That's a good way of phrasing it and we've added that in

**• p. 13, l. 9: "... which is consistent with ..."**

Done

**• p. 14, l. 24: " ... indicating a greater cloud ..."**

Done

**• p. 16, l. 28: "... updraft than in aerosol ..."**

Done

**• p. 18, l. 5: "... measurements of aerosol composition ..."**

Done

**Reviewer #3**

The work describes insights on the low level cloud deck which forms during the Monsoon season over South West Africa. These were derived from aircraft measurements of number concentrations of both cloud droplets and aerosol particles in combination with satellite data and some modeling efforts.

It is generally well written, but it seems that data shown in figures does not always support the conclusions drawn. My major concern is about the conclusion that changes in aerosol particle number concentrations (APNC) do not effect cloud droplet number concentrations (CDNC). And I will not argue that there is a large effect, but the presented data also does not support that there is no effect.

I should say upfront that I started reading this paper not having any agenda of my own on this topic. While writing this report, time and again I saw hints in the manuscript which support an influence of aerosol particle number concentrations (APNC) on cloud droplet number concentrations (CDNC). But a contrasting interpretation was given, arguing that the amount of data for the "low concentrations cases" (i.e., offshore) is too small for interpretations. In the following, I will comment on a number of related text passages and on data shown in your figures which seem to indicate a different result. Your work needs to account for that.

Nevertheless, the topic is timely and a revised version may merit publication in ACP.

We thank the reviewer for their useful comments. We will address them individually below.

**Major concerns:**

As said above, this major concern deals with a possible effect of APNC on CDNC. I understand that you try not to over-interpret the (comparably small number of) data you got offshore. But still, I see the same there, over and over again: Cloud droplet number concentrations (CDNC) are lower offshore than inland, and so are aerosol particle number concentrations (APNC, in your case particles larger than 250nm (N_250). And also the aerosol particle size distribution (APSD) you derived for the modeling is clearly lower in number and different in shape offshore, compared to inland.

If I understood it correctly, you argue that even offshore the air masses are always polluted, originating from biomass burning aerosol that prevails in this region during this time of the year (e.g., page 21, line 24-25). This is a bit confusing, as you also describe that you went > 20km offshore "to unambiguously remove any possibility of local terrestrial emissions affecting" (page 10, line 25-26). So this would mean that the pollution from biomass burning aerosol should be ubiquitous in the whole region, also >

20km offshore. But the size distribution you show as representative for > 20km offshore (Fig. 8) does not indicate that biomass burning pollution is prominently present (for that a pronounced mode of particles < 100nm would be expected). So I somewhat wonder how "polluted" air masses in the offshore region really are.

**Generally, you omit the lower concentrations offshore and if you refer to them at all, you say that these data are not to be trusted, due to their scarcity. It is true that they are scarce, but if you really do not trust them, you should not use them in the first place. This is NOT what I would recommend. But I would still recommend that you spend some thoughts on the idea that there might be a real increase in APNC as the air masses go over land, and related to that an increase in CDCN. And I also recommend that you adjust your interpretations in the text.**

We thank the reviewer for their in-depth and detailed comments. The preceding set of paragraphs cover several aspects and are difficult to divide up, so we will try to address them as a group. Firstly, the reviewers comment about the biomass burning size distribution does not apply to biomass burning that has undergone any sort of ageing, and literature shows aged biomass burning is primarily in the accumulation mode and consistent with our offshore measurements (e.g. Haywood et al., 2003; Sakamoto et al., 2014) . It sounds like the reviewer is confused about the source of the biomass burning- it is long-range transported from fires in central Africa- we have said

*"Haslett et al. (2019) recently described the aerosol properties measured during the DACCIWA aircraft campaign, and showed that a large background of transported biomass burning pollution from the southern hemisphere was ubiquitous in the West African boundary layer"*

The idea of going >20km offshore was to unambiguously remove any local component to demonstrate clearly that there was indeed a ubiquitous background of long-range transported pollution across the whole offshore region.

The final point refers to the offshore CDNC measurements. It is not clear what the reviewer is referring to by saying "you omit the lower concentrations offshore" as they are included in many plots and are discussed in numerous places throughout the manuscript. Reviewer #1 has said we should remove them entirely as there's not enough data, but this reviewer has suggested they are either very important or we could remove them if we don't trust them. We do trust them, however we acknowledge they are limited in number.

The main issue is whether the local aerosols are having a large effect on the cloud properties. We have shown that there is no large effect on the broad-scale cloud field downwind of major cities (Section 3.2), and the thick city plumes are a small fraction of the dataset (Section 3.3). With the limited offshore measurements the best way to assess the average effect of local aerosols (as a whole, not just in the city plumes) is by the modelling approach in Section 4.

Having said that, the reviewer's comment has highlighted that it would be useful to state more clearly why the modelling section is there and why this is the best way to address this problem. We have made several changes in response to this comment and ones further down.

In the abstract:

[revised manuscript text omitted]

**Below I collected the passages in your manuscript I stumbled upon, which all more or less deal with this issue:**

**On page 10, line 4-5, referring to Fig. 3, you say "Despite the presence of urban emissions, anthropogenic aerosols had no clear effect on cloud cover downwind of the major cities,...". This seems vague - by simply looking at Fig. 3, one region with the highest LLC fraction (in the middle of the figure, longitude 3-4°, latitude 8-9°) is north of the biggest city you indicated (LA) and also close to the highest "density" of cities on the map (IB, OG, IL). And then there is another high LLC fraction north of AJ, another larger city. - At the same time, there are mountain slopes, so you might be correct, but judging from the figure you cannot say that anthropogenic aerosol did not have a clear effect. (I wrote this remark given above before reading on, and later found moreon this:) You do show later on that there is an effect (Fig. 4d) on CDNC (which is, on average, increased to~1.5 times higher values, page 12, line 2), and you again show this in your modelling – so there might not be a large effect, but still, there is one, so your remark on page 10, line 4-5 should be tuned down, maybe already pointing towards these later results, or deleted at all. And I think it is all the more interesting that you see this increase in the CDND in Fig. 4d ALTHOUGH "However, these highly polluted plumes represent less than 10% of the inland data measured over the region."(as you say on page 12, lin 2-3).**

The reviewer does not seem to be clear on the wind direction. We have added this to Figure 3 to show the range at this time of day. Looking at the area the reviewer says is downwind of Lagos (LA), much of it is actually not downwind of Lagos. It is arguably far downwind of Lome (LO) and Cotonou (CO), but if we imagine a plume travelling downwind from these cities, the cloud cover only increases when you reach the hills. Most of the area they say is downwind of Abidjan (AJ) is actually not downwind of Abidjan. Bearing the wind direction in mind, the reviewer's argument doesn't really stand up to scrutiny. The part about CDNC being 1.5x higher inland refers only to the thickest city plumes, and was not widespread. This is already stated in the manuscript. Please also see our response to Rev#2 starting "We have added to section 3.2"

**page 11, line 10: Already upon my first reading, I made the following note here, and this got only more relevant: The most striking difference between offshore and inland data, as I see it, is the larger average CDNC for the inland data. You do not discuss this at all. Would there be generally larger aerosol**

**concentrations over land (also from natural sources)? Or any other idea where that could come from?**

Please see our response to R3 Major concerns, and our changes to the discussion

**page 12, line 14-15: Here you clearly describe that the ground acts as an aerosol source: "Inland, accumulation mode aerosol concentrations were highest closer to the ground, and decreased with altitude up to around 2 – 2.5 km." There is a factor of roughly 2.5 between values of N_250 on ground and above 2km.**

Agreed, this is a good insight. We have added to Sect. 3.4

*"Together, these aerosol profiles suggest local terrestrial sources adding to the background aerosol on a regional scale."*

**page 13, line 1-2: When looking at Fig. 5, it struck me that up to 2.5km mostly N_250 and CDNC are roughly similar for the offshore cases, while for the inland cases CDNC are roughly double the amount of N_250. That would imply that there must be more particles < 250nm in the air inland (compared to offshore) which can be activated to cloud droplets, otherwise the CDNC are difficult to explain.**

We agree it is difficult to explain using aerosol alone. We performed an extensive set of model runs to investigate the relative impacts of the offshore and local aerosol in clouds with different dynamics. Please see the updated discussion, which addresses this issue. We also agree that there are differences in the aerosols smaller than 250nm, as we have shown in Fig. 8. This is discussed further by Haslett et al. (2019).

**page 13, line 14ff: You say "There are differences in aerosol and CDNC between inland and offshore clouds, but not in R_Eff, : : : " – below 1.5km, the "bump" that shows up in CDNC for the offshore case (towards lower values at 0.75 km and 1.25 km) is reflected in larger R_Eff (comparing _ 8-9 micrometer offshore to _ 6-7 micrometer inland). So this sentence needs a reformulation. Already Fig. 4b showed a totally different cloud droplet size distribution for offshore and inland conditions.**

In the preceding sentence, we have already said

*"Between 0.5 and 1.5 km, the clouds offshore tended to have higher $R_{Eff}$ than those measured inland at the same altitude…."*

Figure 4b does not show a cloud droplet size distribution, but we have already addressed the different PDFs of CDNC, that it does show, multiple times in the results and discussion.

**page 14, line 2-3: There is a factor of _ 1.5 between N_250 that you give for offshore and inland, again with concentrations inland being larger. This, together with my comment concerning page 13, line 1-2, I am not sure if it can really be said that "aerosol imported from offshore comprised the majority of accumulation mode aerosol inland".**

There would need to be a factor of >2 increase inland in order for the majority of aerosol to not be from offshore. The local contribution would have to be larger than the transported pollution, which our measurements show it is not.

**page 15, line 8-10: Increase in mean updraft and in CDNC do not really mirror each other. While CDNC increase from 7 to 13 LST and then stay quite stable, the vertical velocity, implying a possibly stronger role of dynamics, is quite stable and has a distinct peak at 17 LST (followed by data ta 15 and 19 LST). So it is not so clear to me that dynamics explain the observations.**

We have changed this to say

*"….suggest that dynamics played a role in the diurnal cycle of CDNC, but cannot fully explain the changes."*

**page 15, line 23-25: It would be good to know how the inland size distribution exactly was derived (measured when and where)? But in any case, here it can clearly be seen that inland and offshore particle number size distributions differ massively, with much higher concentrations for particle sizes _ < 200 nm for the inland aerosol.**

Please see our response to Rev #1's comment starting "Generally aerosol size distributions…"

**page 19, line 21-25: As said above, looking at your data gave me an impression that is deviating from your interpretation. Instead, and increase in particle number concentrations from offshore to inland conditions might indeed play a role.**

Our modelling simulations agree that the local pollution played a role, and we have said as much in the abstract, results, discussion and conclusions.

**page 19, line 21-23: "It may be that our poor statistics of offshore data present a misleading distribution in Fig. 4b, or that there was a systematic and ubiquitous increase in CDNC due to some other factor such as stronger updrafts inland." Again, the "poor statistics" offshore is certainly better constrained than stronger updrafts inland. Admittedly, an increase by a factor of 1.5 for CDNC might not be huge, but at least there seems to be some effect, there.**

Please see our earlier comment regarding changes to the discussion

**page 20, line 23-24: In light of all the above, this sentence should be revised.**

This refers to the broad-scale cloud field, please refer to our comment starting "The reviewer does not seem to be clear on the wind direction…"

**page 21, line 1-3: And again, there is evidence in your data that APNC increased as soon as air masses were transported over land, although they might have been already "mildly polluted". (The particle number size distributions offshore and inland are VERY different!) And it is true that polluted air masses are less susceptible to a further increase in APNC, but you do see an increase, nevertheless. So again, this sentence should be revised.**

We have changed "limited susceptibility" to "reduced susceptibility". They do have reduced susceptibility because doubling the local aerosol only causes a 13% increase in CDNC, not a ~100% increase. We have replaced similar terms like "limited effect" with "reduced effect" throughout the paper

**page 21, line 24-25: Again, the "high aerosol background from transported smoke" was somewhat lower than what that sentence suggests.**

We have changed it to just "the aerosol background"

**Minor and technical comments:**

**page 9, line 11-13: You write "A further region of higher cloud cover was seen just inland : : : ". If I get the color scale right, then closer to the coast the coverage is 70%, drops to 60% and then again increases to 70%. The description here sounds a bit different. To me this area does not necessarily look like two separate regions, and the clouds closer to the sea do not really dissipate as they move further inland. A reformulation is needed.**

The inland region starts forming at 3AM, but the sea breeze clouds start forming around 9AM, so they are different. There's not necessarily a full gap between them at midday in terms of total cloud cover, but the two bands show different daily cycles.

**page 13, line 6: "is" is missing between "This due".**

Done

**page 13, line 12: The sentence is more general than what you report. Following "This implies that offshore clouds" insert "at altitudes above 1.5 km" and in the following line change "that clouds offshore" to "that these offshore clouds".**

Done

**page 14, line 18-19: The CDNC does not really "peak around 15:00 LST". There is a clear increase from before to after local noon, but after that, looking at all mean and percentiles, it seems to rather reach a plateau, until after 18:00.**

This now says

*"A trend is apparent in the data, as CDNC increased inland throughout the morning and remained higher through the afternoon"*

**page 17, line 1-2: You say ": : : for the average updrafts the main sensitivity of CDNC is not so clear-cut." What do you mean by "is not so clear-cut"? For the average updrafts, there is a difference, as you state in the text above, so I am confused by this statement here.**

We have replaced this with

*"…the clouds are in a regime where they are sensitive to both changes in updraft and aerosol"*

**page 17, line 3-4: You say that "A large fraction of the day-to-day variability : : : derives from the day-to-day variability in the offshore aerosol brought inland." But isn't this indicated by the bluish color band? The variability of the inland aerosol (greyish color band) is much larger. This confused me a little (I thought you might have mixed up the colors, but maybe you didn't?), and it would be fair to at least mention the fact that the inland aerosols' variability is even larger.**

We have added a sentence to clarify

*"At higher updrafts, the variability from offshore aerosol causes ~50% of this inland variability, but at lower updrafts this fraction increases to 100%."*

**page 17, line 4: Following the above mentioned sentence, you mention "these average updrafts", and maybe I was still confused from that "color issue"**

**above, but I first didn't get which updrafts you mean. Maybe it would be better to relate to them as "the average morning and afternoon updrafts as defined above", or something like this.**

Done

**page 17, line 18-19: The change in CDNC you report for omitting local emissions in your model also nicely reflects the change in CDNC between offshore and inland that you showed above. You should mention this.**

We have addressed this issue in the updated discussion

**page 17, line 20ff: "The effect of varying local aerosol emissions does not produce a linear relationship between CDNC and total aerosol concentrations." One additional reason for this, which you do not mention explicitly, yet, is that there is this rivalling effect where the maximum super-saturation at a given updraft is lower when there are higher concentrations of CCN, due to the water vapor being consumed faster, hence the critical size above which particles activate to droplets becomes larger.**

We have added

*"This nonlinearity may be related to a reduction in supersaturation for a given updraft velocity, as CCN number concentration increases and water vapour condenses more quickly."*

**page 18, line 21: Is "flat" a good way to describe an ocean surface? Maybe it is (I'm not a native speaker), but I would think there might be a better word, like "calm" or something else?**

We have clarified *"flat (compared to the land surface)"*

**page 22, line 22: There are too many quotation marks around "borderline".**

Fixed

**References not in the main text**

Haslett, S. L., Taylor, J. W., Evans, M., Morris, E., Vogel, B., Dajuma, A., Brito, J., Batenburg, A. M., Borrmann, S., Schneider, J., Schulz, C., Denjean, C., Bourrianne, T., Knippertz, P., Dupuy, R., Schwarzenböck, A., Sauer, D., Flamant, C., Dorsey, J., Crawford, I. and Coe, H.: Remote biomass burning dominates southern West African air pollution during the monsoon, Atmos. Chem. Phys. Discuss., 1–23, doi:10.5194/acp-2019-38, 2019.

Haywood, J. M., Osborne, S. R., Francis, P. N., Keil, A., Formenti, P., Andreae, M. O. and Kaye, P. H.: The mean physical and optical properties of regional haze dominated by biomass burning aerosol measured from the C-130 aircraft during SAFARI 2000, J. Geophys. Res. Atmos., 108(D13), doi:10.1029/2002jd002226, 2003.

Sakamoto, K. M., Allan, J. D., Coe, H., Taylor, J. W., Duck, T. J. and Pierce, J. R.: Aged boreal biomass burning aerosol size distributions from BORTAS 2011, Atmos. Chem. Phys. Discuss., 14(17), 24349–24385, doi:10.5194/acpd-14-24349-2014, 2014.

---

## Author Response (AR2)

We thank the two Reviewers for taking the time to re-review our revised manuscript, and the Editor for overseeing the review process.

**Reviewer #1**

**The authors have thoroughly adjusted the manuscript based on the comments of the three reviewers. I believe most of the changes and responses are satisfactory and the manuscript is improved. I would like to suggest one minor revision to the wording of one of their main points. If my understanding is correct, the authors assume that the onshore particles greater than 250 nm is equal to the particles offshore + particles with onshore sources. For example, if offshore >250 nm concentrations were observed to be 100 cm^-3 and on shore was observed to be 150 cm^-3 the assumption the authors use indicates that 100 of the 150 particles on shore are from long range transport offshore.**
**I am not so sure I can agree with this assumption. At least not without some sort of tracer to identify if the transported air mass has become diluted. The authors do use CO as an indicator of offshore long range transport of biomass burning, however CO onshore would also be influenced by local sources making it difficult to identify the contribution of offshore particles to onshore concentrations. Even if one were to assume the on shore sources of CO had little effect on most of the measurements (probably a bad assumption), an onshore CO concentration lower than the offshore CO concentration would indicate dilution of the airmass and suggest fewer particles are from long range transport. I am not sure how much, but I would expect there to be some dilution of particles coming onshore.**
**I suggest a more general statement like: onshore accumulation mode particles are only 40% higher than offshore concentrations (rather than assuming most onshore particles originated from offshore as you have stated: "aerosol imported from offshore comprised the majority of accumulation mode aerosol inland" ). This difference in accumulation mode particles, along with differences in dynamics, accounts for the mildly higher (~50%) CDNC onshore.**

From the lead author: This makes sense now, I understand the point the reviewer was trying to make in the previous comments. The way I was thinking about it was like a conveyor belt moving marine air inland. It's not clear how you would get significant horizontal dilution if everything is moving in a similar manner, but I can see how you could get some dilution vertically through increased convection inland, and also there will have been some deposition of the marine aerosol, so you are right that we haven't demonstrated that it's the same particles.

From all authors: We have changed Section 3.5 to now say

*"Together with the good correlation, this suggests that aerosol imported from offshore had a strong influence on the average amount, and day-to-day variability, of accumulation mode aerosol inland"*

And in the abstract is now says *"A significant fraction of…"* rather than *"The majority of…"*, and the same to Section 4.1.

In the discussion it now says

*"In localised city plumes, highly concentrated emissions had a noticeable increase in CDNC, but these concentrated plumes were not widely observed over the regional scale."*

**Reviewer #2**

The manuscript has improved significantly and the authours managed to make things much clearer with the revisions they made. Issues I had with the text are now removed. Therefore I recommend publication in ACP now!

Technical comments (to the version with the "tracked changes" in it):

1) page 5, line 25: "Aerosol data were screened for cloud using a threshold for LWC of ..." - and I guess those data taken within clouds were excluded - you may want to add this bit of information explicitily.

Yes, we have clarified they were removed

2) page 15, line 18: "The ubiquitous presence aged biomass burning smoke transported ...". Is an "of" missing between "presence" and "aged"?

Yes, done!

3) page 16, line 12: "and this reduces the sensitivity of CDNC to local emissions, which form less of the total." Are the last 6 words a leftover and should be deleted? Or "the total of WHAT"?

We've added "…the total aerosol."

4) I know that you exchanged "limited" with "reduced" in some parts of the text based on comments I made in my first review (mainly the "Conclusion"), but I'd be fine if you used "limited" again, instead ("reduced" always makes me wonder "reduced with respect to what") - but I leave this up to you, if you prefer "limited" or "reduced" in these locations.

In the conclusions the sentence starts "Compared to clean conditions……" so that should be clear. We've added in the discussion "…
[revised manuscript text omitted]